# Interactions between circuit architecture and plasticity in a closed-loop cerebellar system

Hannah L Payne[1], Jennifer L Raymond[2]*, Mark S Goldman[3,4]*

[1]Zuckerman Mind Brain Behavior Institute, Columbia University, New York, United States; [2]Department of Neurobiology, Stanford University, Stanford, United States; [3]Center for Neuroscience, Department of Neurobiology, Physiology and Behavior, University of California, Davis, Davis, United States; [4]Department of Ophthalmology and Vision Science, University of California, Davis, Davis, United States

**Abstract** Determining the sites and directions of plasticity underlying changes in neural activity and behavior is critical for understanding mechanisms of learning. Identifying such plasticity from neural recording data can be challenging due to feedback pathways that impede reasoning about cause and effect. We studied interactions between feedback, neural activity, and plasticity in the context of a closed-loop motor learning task for which there is disagreement about the loci and directions of plasticity: vestibulo-ocular reflex learning. We constructed a set of circuit models that differed in the strength of their recurrent feedback, from no feedback to very strong feedback. Despite these differences, each model successfully fit a large set of neural and behavioral data. However, the patterns of plasticity predicted by the models fundamentally differed, with the direction of plasticity at a key site changing from depression to potentiation as feedback strength increased. Guided by our analysis, we suggest how such models can be experimentally disambiguated. Our results address a long-standing debate regarding cerebellum-dependent motor learning, suggesting a reconciliation in which learning-related changes in the strength of synaptic inputs to Purkinje cells are compatible with seemingly oppositely directed changes in Purkinje cell spiking activity. More broadly, these results demonstrate how changes in neural activity over learning can appear to contradict the sign of the underlying plasticity when either internal feedback or feedback through the environment is present.

*For correspondence:
jenr@stanford.edu (JLR);
msgoldman@ucdavis.edu (MSG)

## Editor's evaluation

Payne et al. present a novel model that predicts sites and directions of plasticity within the vestibular cerebellum to explain the basis for learned adjustments to reflexive eye movements in monkeys. The model is convincingly constrained by prior biological observations and makes an important prediction about the level of feedback available to the cerebellar cortex and how this level determines the plasticity required to explain post-learning changes in activity. Overall, a number of exciting and testable experiments will likely be motivated by this study.

## Introduction

Synaptic plasticity distributed across multiple sites of a circuit is thought to underlie changes in behavior. To understand how such plasticity supports learning, it is necessary to identify sites of plasticity, determine how plasticity at these sites produces changes in neural activity, and link these changes to behavior. Thus far, such characterization has proven difficult in the vertebrate brain.

**Figure 1.** Feedback can obscure plasticity in neural systems. (**A**) Purely feedforward circuit. A decreased neural response to a stimulus after learning can be attributed to LTD of excitatory synapses (or equivalently, LTP of inhibitory synapses) upstream of the recorded neuron. *Triangles*, plastic synapses. (**B**) Recurrent circuit, with both internal and external feedback. The same decreased neural response can no longer be definitively attributed to plasticity at nominally upstream sites. Instead, plasticity at inputs to a second site that is a postsynaptic target of the recorded neuron, and thus appears to be downstream, may feed back to affect the recorded neuron's activity. Such 'downstream' plasticity may mask the effects of 'upstream' plasticity. (**C**) Feedforward circuit proposed by the Marr-Albus-Ito model to support cerebellum-dependent motor learning. Before learning, vestibular (head velocity) inputs to the brainstem drive compensatory eye movements that are directed opposite to rotation of the head. After learning, LTD of vestibular inputs to Purkinje cells reduces inhibition onto brainstem neurons, increasing eye movement amplitude. (**D**) Simplified version of the feedback circuit proposed by the Miles-Lisberger model. After learning, LTP of vestibular inputs to the brainstem drives larger contraversive eye movements during the VOR in the dark. Efference copy of these eye velocity commands (*red pathway*) leads to decreased Purkinje cell activity, despite LTP of the vestibular input to Purkinje cells.

A key challenge is the ubiquity of feedback loops in neural systems. Feedback loops can be internal to the brain — as in the recurrent circuits thought to underlie short-term memory, predictive coding, and gain control (*Constantinidis and Wang, 2004*; *Douglas and Martin, 2007*; *Fang et al., 2023*) — or partially external to the brain, arising whenever a behavior influences sensory input from the environment that, in turn, influences subsequent behavior, such as during control of movements (*Robinson, 1965*; *Todorov and Jordan, 2002*; *Holst and Mittelstaedt, 1950*). Such feedback loops make it challenging to identify the sites of plasticity that underlie learning. Because direct measurement of synaptic strength is extremely difficult in intact behaving animals, a common approach is to instead infer changes in synaptic strength from observed changes in neural firing (e.g. *Gilbert and Thach, 1977*; *Moita et al., 2003*; *Yao and Dan, 2001*). As a simplifying assumption, neural systems are commonly treated as if they are feedforward circuits, with changes in neural activity at a particular site attributed to plasticity somewhere upstream of that site (*Figure 1A*). However, in systems with feedback loops, the distinction between upstream and downstream is ill-defined, so that changes in the activity of a neuron after learning do not necessarily reflect plasticity in its nominally upstream inputs (*Figure 1B*). Thus, feedback loops can confound the inference of changes in synaptic strength from observed changes in neural activity.

Such interactions between circuit feedback and plasticity are at the heart of a decades-long debate about how the cerebellum implements motor learning. The classic Marr-Albus-Ito model assumes a feedforward architecture in which errors are reduced through changes in the synaptic inputs to

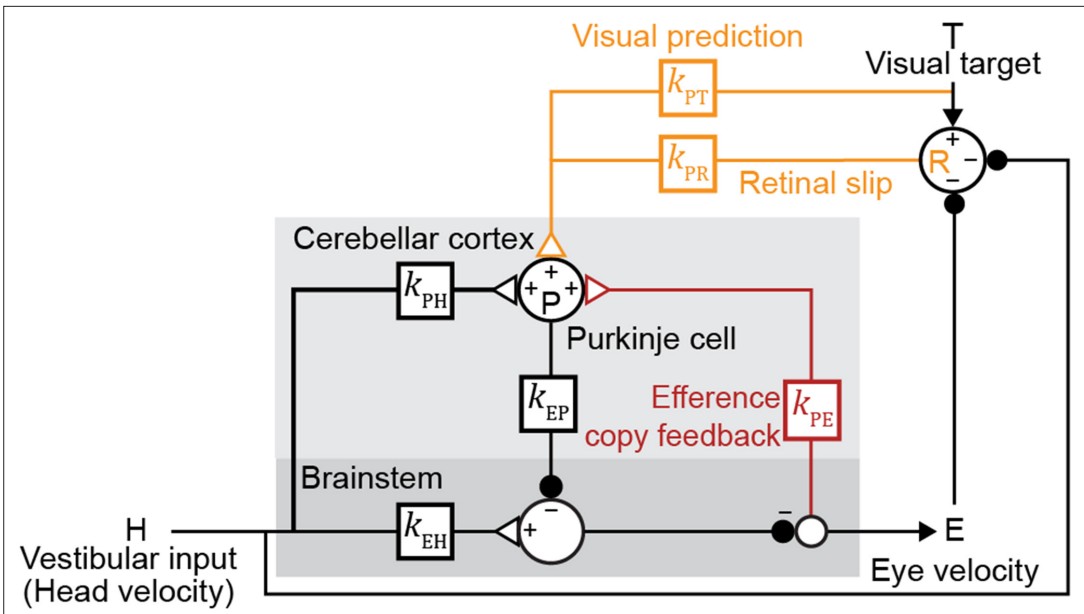

**Figure 2.** Linear filter circuit model. Signal transformations between different nodes of the circuit are modeled as linear filters $k_{XY}$ (boxes). Purkinje cell population firing rate (**P**) is driven by vestibular stimulation (head velocity, **H**) through $k_{PH}$, by efference copy of eye movement commands (**E**) through $k_{PE}$ (red), and by visual signals through $k_{PR}$ (orange, retinal slip velocity) and $k_{PT}$ (orange, predicted visual target velocity). Eye velocity is driven by a direct pathway carrying head velocity input to the brainstem ($k_{EH}$) combined with inhibition from Purkinje cells ($k_{EP}$). The neural circuits between the brainstem and the eye muscles, which compensate for the dynamics of the eye plant, are implicitly included in the filters $k_{EP}$ and $k_{EH}$. For each fixed strength of $k_{PE}$, all other filters were fit to the data.

Purkinje cells, the sole output neurons of the cerebellar cortex (*Figure 1C*; *Marr, 1969*; *Albus, 1971*; *Ito and Kano, 1982*). This is consistent with a large number of studies suggesting that long-term depression (LTD) occurs at the excitatory parallel fiber synapses onto Purkinje cells in response to error signals carried by climbing fiber inputs, effectively implementing reinforcement learning through error-driven plasticity ('parallel fiber-Purkinje cell LTD'; *Coesmans et al., 2004*; *Gilbert and Thach, 1977*; *Ito and Kano, 1982*; *Kimpo et al., 2014*; *Medina and Lisberger, 2008*; *Sakurai, 1987*; *Silva et al., 2023*; *Yang and Lisberger, 2013*; *Yang and Lisberger, 2014*, but see *Schonewille et al., 2011*). In contrast, later experimental observations raised the possibility that the learning-related changes in Purkinje cell firing could instead be due to feedback of changes occurring *outside* of the cerebellar cortex ('Miles-Lisberger model', *Figure 1D*; *Hirata and Highstein, 2001*; *Lisberger, 1994a*; *Lisberger et al., 1994c*; *Lisberger et al., 1994b*; *Miles and Lisberger, 1981*). Furthermore, these experiments were interpreted as evidence that parallel fiber-Purkinje cell plasticity within the cerebellar cortex was in the opposite direction (long-term potentiation, LTP) from the LTD predicted by the Marr-Albus-Ito model. These opposing conclusions about the sites and directions of plasticity underlying cerebellum-dependent motor learning have remained unreconciled for decades. In particular, it was not clear whether a model consistent with parallel fiber-Purkinje cell LTD could also explain the full suite of experimental observations that motivated the Miles-Lisberger model.

Here, we use a data-driven, computational approach to determine how the strength of feedback in a circuit determines the sites and directions of synaptic plasticity required to accomplish a given change in neural output and behavior. We aggregated neural and behavioral data from a large set of experiments testing oculomotor performance and learning, and fit a series of computational models that systematically differed in the strength of internal feedback. We find that models with weak or no internal feedback are consistent with climbing fiber-driven LTD at parallel fiber-Purkinje cell synapses and explain all experimental observations, including paradoxical changes in neural activity during a closed-loop visual task that appear to contradict the underlying plasticity. Our results provide a solution to a longstanding debate in the cerebellar field, and more broadly demonstrate how, in

closed-loop systems, synaptic strength and neural activity at a given site can, counter-intuitively, change in opposite directions.

## Results

### Oculomotor learning circuit and closed-loop modeling strategy

We perform our studies within the context of the learned control of eye movements. Oculomotor learning provides a powerful experimental system because it regulates a relatively simple transformation from sensory inputs to motor output in a circuit whose anatomy and physiology have been extensively characterized. The oculomotor system generates eye movements to stabilize visual images on the retina during both motion of the body and motion of visual objects in the world. Vestibularly driven eye movements, known as the vestibulo-ocular reflex (VOR), stabilize gaze by counter-rotating the eyes in response to head movement and occur even in complete darkness. Visually driven eye movements, including 'smooth pursuit' eye movements that track a moving target, stabilize images on the retina in response to visual input (**Noda and Suzuki, 1979**; **Rambold et al., 2002**).

Both the visual and vestibular functions of the oculomotor system are remarkably linear (**Bagnall et al., 2008**; **du Lac and Lisberger, 1995**; **Lisberger and Fuchs, 1978**; **McElvain et al., 2015**; **Payne et al., 2019**; **Walter and Khodakhah, 2006**). Hence, we modeled this circuit with a network of linear temporal filters, $k_{XY}$, each representing the transformation of a signal over a (mono- or multi-synaptic) anatomical neural pathway to node X from node Y (**Figure 2**). Since a given pathway may contain both excitatory and inhibitory neurons, the net contribution of the excitatory and inhibitory synapses within each pathway is represented by a single linear filter that can have positive or negative value at any given time point. Vestibular sensory input encoding angular head velocity (H) drives eye movements (E) via a 'direct pathway' through brainstem nuclei ($k_{EH}$; **Figure 2**, *black*), and an indirect side loop through Purkinje cells (P) in the floccular complex of the cerebellar cortex ($k_{PH}$; **Figure 2**, *black*; **Voogd et al., 2012**). Purkinje cells also receive efference copy signals related to eye movement commands ($k_{PE}$; **Figure 2**, *red*) and visual signals related to image motion (**Voogd et al., 2012**). The visual pathway is subdivided into a 'retinal slip' (R) pathway conveying motion of images across the retina with a delay ($k_{PR}$; **Figure 2**, *orange*), and a 'visual prediction' (T) pathway providing non-delayed information about predictable target motion ($k_{PT}$; **Figure 2**, *orange*). While not needed to explain the main qualitative results of our paper, the visual prediction pathway was included to model oculomotor tracking of predictable visual targets with no delay or in the absence of sustained retinal slip (**Figure 3—figure supplement 1**; **Becker and Fuchs, 1985**; **Kowler and Steinman, 1979**; **Leung and Kettner, 1997**; **Stone and Lisberger, 1990**; reviewed in **Kowler et al., 2019**). Such visual prediction signals have been recorded in cortical pathways that provide input to the floccular complex of the cerebellar cortex (**Ilg and Thier, 2008**) and are incorporated in previous models of smooth pursuit (**Barnes and Asselman, 1991**; **Kowler et al., 1984**; **Orban de Xivry et al., 2013**). Finally, neurons in the vestibular nucleus of the brainstem combine inhibitory input from the Purkinje cells ($k_{EP}$; **Figure 2**, *black*) with direct vestibular input ($k_{EH}$; **Figure 2**, *black*), and project to motor neurons in the abducens and oculomotor nuclei to control eye velocity.

Understanding how changes in each of these pathways contributes to learning requires identifying the signal transformations occurring in each pathway. This is challenging because the vestibular, visual, and efference copy signals are tightly correlated due to feedforward and feedback interactions. Previous models have attempted to address this issue by assuming a particular strength of efference copy feedback, or have quantitatively fit simpler open-loop models to limited sets of data that may not fully eliminate the confounds stemming from strongly correlated predictor variables (**Blazquez et al., 2003**; **Hirata and Highstein, 2001**; **Lisberger, 1994a**; **Tabata et al., 2002**). Here, we fit an extensive set of Purkinje cell and eye movement data recorded in monkeys before and after learning, while systematically varying the strength of efference copy feedback (by setting the strength of filter $k_{PE}$) to separate the contributions of different, correlated pathways and enable solutions that have not previously been considered (see Materials and methods). To infer plastic changes in the circuit, we then compared the inferred filters before and after learning.

## Before learning: degeneracy of model fits

Linear filter models with any level of efference copy input, ranging from a feedback gain of 0 ('No Feedback') to 1 ('Strong Feedback'), closely accounted for the dynamics of neural activity and behavior before learning (*Figure 3*). The data shown in *Figure 3* consist of Purkinje cell and eye velocity responses to 25 different oculomotor stimulus conditions, including vestibular input alone (VOR), visual input alone (smooth pursuit), and combinations of visual and vestibular input, delivered as sinusoids or steps of stimulus velocity. Responses during additional sinusoidal frequencies of the VOR in the dark before learning were also included from datasets that include learning (see section 'After learning: circuit feedback affects inferences about plasticity' and Materials and methods). For every stimulus condition, both the No Feedback and Strong Feedback models fit the data similarly (*Figure 3—figure supplement 2*). Models with negative internal feedback also fit similarly (not shown), whereas positive feedback gains greater than one were not considered since such feedback causes instability. The strength of efference copy feedback ($k_{PE}$) to Purkinje cells was thus unconstrained by a large set of oculomotor data before learning.

To understand how models with vastly different efference copy feedback strengths could produce such similar outputs, we examined how other filters changed to compensate for the changing level of feedback. Some filters did not depend on feedback strength: the filters conveying vestibular input and Purkinje cell activity to the brainstem, $k_{EH}$ and $k_{EP}$, were nearly identical in all models, indicating that these pathways are well constrained by the data regardless of the strength of efference copy feedback (*Figure 4A*). By contrast, the filters carrying vestibular and visual inputs to Purkinje cells did vary with feedback strength. The vestibular input filter to Purkinje cells, $k_{PH}$, changed from small and net negative in the No Feedback model to large and net positive in the Strong Feedback model (*Figure 4B*, *top*). Here, a positive filter weight indicates a net excitatory effect from an ipsiversive stimulus (e.g. increased excitation of a Purkinje cell in the right cerebellar hemisphere during head rotation to the right) and an inhibitory effect of a contraversive stimulus, whereas a negative filter weight indicates the opposite. The small net amplitude of the $k_{PH}$ filter in the No Feedback model, evidenced by the small steady state step response (*Figure 4B*, *right*), directly reflects the experimental observation that Purkinje cells have minimal modulation of firing rate during the VOR. By contrast, in the Strong Feedback model, the same minimal modulation is achieved by a net positive $k_{PH}$ filter that offsets the negative efference copy input through $k_{PE}$. The retinal slip filter also varied with feedback strength, changing from acceleration-like to velocity-like as feedback strength decreased (*Figure 4B*, *bottom*; see schematics in *Figure 4C*)—this reflects that, in the Strong Feedback model, the efference copy feedback loop forms a temporal integrator that converts acceleration-like inputs to velocity-like outputs. The inferred strength and dynamics of the vestibular and visual inputs to the cerebellar cortex therefore are not well-constrained by this extensive set of data, and varied depending on the assumed strength of efference copy feedback, even before learning.

To understand the source of this apparent degeneracy, we analyzed a slightly simplified model with a single combined visual pathway that permits the analytic calculation of a closed-form solution for the model equations (Materials and methods). This analysis revealed a strict degeneracy for some, but not all, parameters in the model. This is visualized in *Figure 5* for the steady-state component of the response by plotting the model cost function as the relevant filter parameters were varied. Consistent with the results of *Figure 4A*, the two brainstem pathways, $k_{EH}$ and $k_{EP}$, were fully constrained, as illustrated by the single minimum in the cost function landscape (*Figure 5A*). By contrast, there was a degenerate direction in parameter space for the three inputs to Purkinje cells: $k_{PH}$, $k_{PE}$, and $k_{PR}$, as illustrated by the flat valley in the cost function landscape, where the fit to the data was equally good for a range of different filter strengths (*Figure 5B*). In this degenerate direction, $k_{PH}$ and $k_{PE}$ increase together while $k_{PR}$ decreases. As discussed above, the concurrent increase in $k_{PH}$ and $k_{PE}$ indicates that the small Purkinje cell responses observed during the VOR in the dark could reflect either a small vestibular input ($k_{PH}$) alone, or a large vestibular input offset by sufficient efference copy feedback ($k_{PE}$). However, our analysis shows that the degeneracy is actually between all three (vestibular, efference copy, and visual) pathways rather than just the vestibular and efference copy feedback pathways. Ultimately, this degeneracy reflects that there are only two independently controllable variables, the vestibular and visual target stimuli, yet three unknown input filters. Eye velocity is not an independent variable, because it is determined by how these stimuli are processed by the circuit. In contrast, there are only two unknown input filters to the brainstem, and thus they are both fully constrained. The

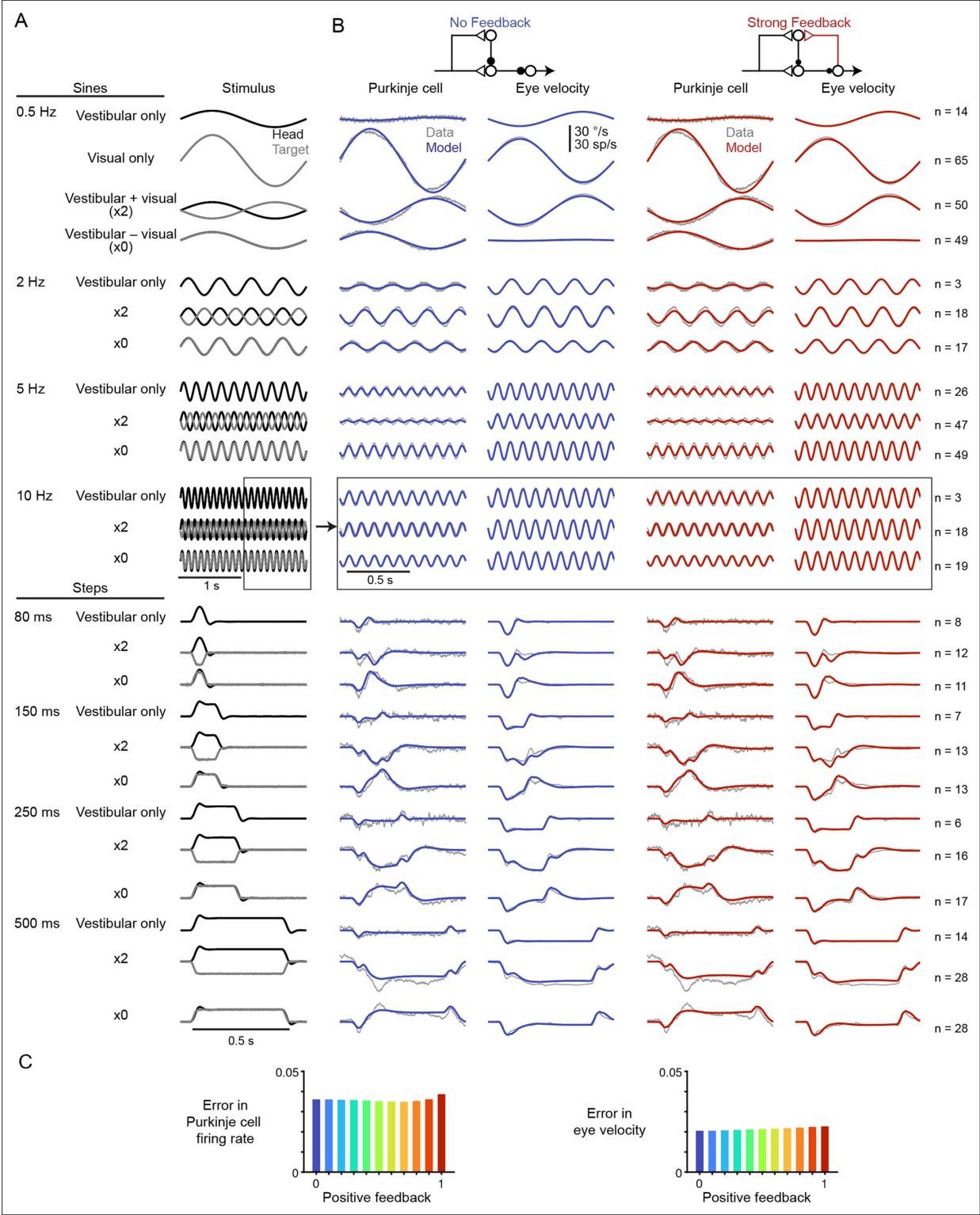

**Figure 3.** Models with or without efference copy feedback fit neural and behavioral data before learning. (**A**) Vestibular ('Head', *black*) and visual ('Target', *gray*) stimuli for each behavioral condition. Conditions consisted of vestibular input alone ('Vestibular only', i.e. VOR in the dark), visual input alone ('Visual only', i.e. smooth pursuit), vestibular input paired with oppositely directed visual input such that eye movements twice as large as normal were required to stabilize the image ('x2'), and vestibular input paired with visual input in the same direction such that eye movements must be eliminated to stabilize the image ('x0', or VOR cancellation). Ipsiversive head and eye movements are plotted as positive values. (**B**) Purkinje cell firing rate and eye velocity measured experimentally (*gray*) and predicted by the No Feedback model (*blue*) or the Strong Feedback model (*red*). Thickness

*Figure 3 continued on next page*

*Figure 3 continued*

of experimental (*gray*) trace indicates standard error of the mean. (**C**) Normalized root mean squared error of fits to Purkinje cell firing rate (*left*) and eye velocity (*right*) for all models.

The online version of this article includes the following figure supplement(s) for figure 3:

**Figure supplement 1.** Model response to occlusion of a moving visual target.

**Figure supplement 2.** Model error for individual behavioral conditions.

practical implication of this analysis is that the vestibular, visual, and efference copy feedback filters to Purkinje cells cannot be fully determined using neural recording and behavioral data alone, but require an additional experimental strategy.

## After learning: circuit feedback affects inferences about plasticity

The fundamental problem of degeneracy before learning applies after learning as well, with critical implications for inferred learning mechanisms. Learning can increase or decrease the amplitude of the eye movements driven by the VOR. For simplicity, we focus below on learned increases in the VOR unless otherwise specified; complementary changes occurred in the model during learned decreases in the VOR (*Figure 6—figure supplements 1 and 2*). We also, for ease of presentation, describe the response of the circuit to ipsiversive head turns (passive head rotation towards the side of the recorded neurons); the same arguments hold for contraversive head turns, but with opposite changes in firing throughout the VOR circuit.

Learned increases in the VOR can be induced by pairing a vestibular stimulus with oppositely directed motion of a visual stimulus so that larger-than-normal eye movements are required to stabilize the image on the retina. Following such learning, Purkinje cell responses to an ipsiversive vestibular stimulus alone (in the dark) *decrease* (*Blazquez et al., 2003*; *Hirata and Highstein, 2001*; *Lisberger et al., 1994b*; *Miles et al., 1980*; *Watanabe, 1985*), which disinhibits brainstem target neurons, thereby increasing the amplitude of eye movements. Whereas these changes in behavior and neural activity are well established, longstanding controversy concerns the sites and directions of plasticity underlying these changes.

The Marr-Albus-Ito model proposes that the observed decrease in Purkinje cell firing is caused by a decrease in the strength of vestibular input to Purkinje cells via LTD of vestibular parallel fiber-Purkinje cell synapses (*Figure 1C*; *Albus, 1971*; *Ito and Kano, 1982*; *Marr, 1969*). However, the Marr-Albus-Ito model does not consider the implications of a potential efference copy feedback pathway to Purkinje cells. In particular, efference copy signals encoding altered eye movements might drive altered Purkinje cell firing (*Lisberger, 1994a*).

Later studies by Miles and Lisberger attempted to isolate the contribution of vestibular input to Purkinje cell firing from any influence of efference copy signaling. They used a behavioral paradigm known as VOR cancellation, in which the eyes track a visual stimulus that moves exactly with the head, thus canceling the normal VOR eye movement response. During this paradigm, eye velocity in the orbit is close to zero — along with, presumably, any associated efference copy signals. Purkinje cell activity during VOR cancellation was thus attributed to vestibular input alone. Surprisingly, Purkinje cell activity during VOR cancellation *increases* after learning — opposite to the *decrease* in activity (*Figure 1C and D*) observed during an identical vestibular stimulus presented in the dark (*Lisberger, 1994a*; *Miles and Lisberger, 1981*). This increase in Purkinje cell activity during VOR cancellation was interpreted as evidence that the vestibular inputs to Purkinje cells must undergo potentiation, rather than depression, during learning (*Miles and Lisberger, 1981*). The decrease in Purkinje cell activity observed during the VOR in the dark was then attributed to plasticity in the brainstem ($k_{EH}$) that is relayed to the cerebellar cortex via efference copy feedback ($k_{PE}$; *Figure 1D*), rather than to LTD of vestibular parallel fiber inputs to Purkinje cells ($k_{PH}$). The Marr-Albus-Ito and Miles-Lisberger models therefore differ fundamentally in the plastic changes that they infer in both the cerebellar cortex (vestibular pathway undergoes LTD vs. LTP, respectively) and in the brainstem (no plasticity vs. plasticity).

To assess these seemingly contradictory hypotheses, we examined the linear filters in our models before and after learning (Materials and methods). Each model was fit to learned changes in behavior during the VOR in the dark across a broad range of stimulus frequencies from 0.5 Hz to 50 Hz (data

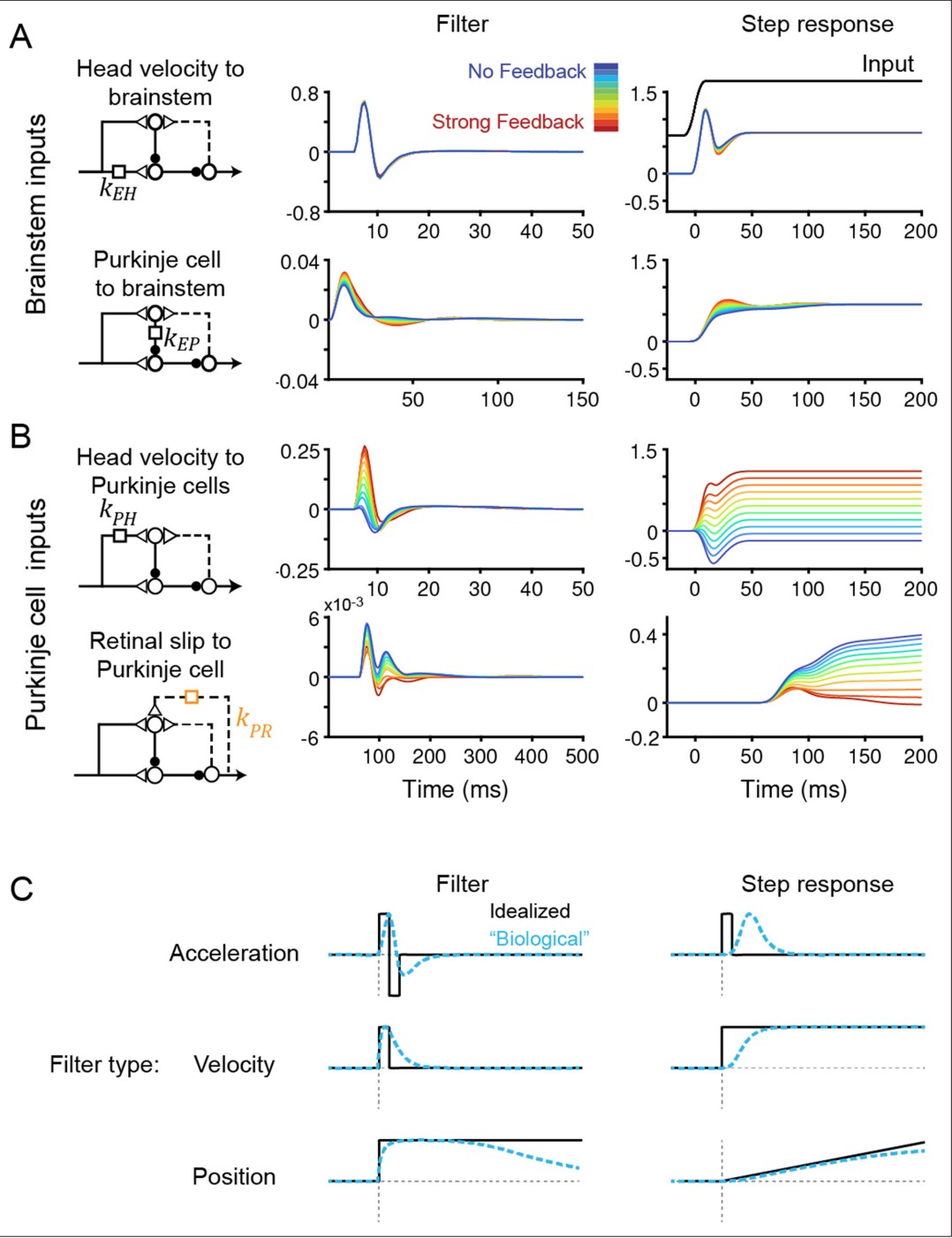

**Figure 4.** Temporal filters before learning are well-constrained for inputs to the brainstem, but not to Purkinje cells. (**A**) Filters conveying head velocity input (*top*, $k_{EH}$) and Purkinje cell activity (*bottom*, $k_{EP}$) to the brainstem for all models, ranging from No Feedback (*blue*) to Strong Feedback (*red*). The actual filter shape (*left*) and the response of the filter to a smoothed 'step' input (*right*) are shown. Following *Equation 1* (Materials and methods), positive weights for $k_{EH}$ cause oppositely directed (negative) changes in eye velocity. Most filters for $k_{EH}$ are hidden beneath the trace for the No Feedback model. Units for step responses: °/s eye per °/s head (*top*), °/s eye per sp/s (*bottom*). For the linear filters, these units are multiplied by s$^{-1}$. (**B**) Filters conveying head velocity (*top*, $k_{PH}$) and retinal slip (*bottom*, $k_{PR}$) input to Purkinje cells, as in (**A**). The filter conveying efference copy input to Purkinje cells ($k_{PE}$) was not fit to data and therefore is not shown here; it is given by an exponential filter of time constant 3 ms, with amplitude scaled according to the stated strength of efference copy feedback (see Materials and methods). Units for step responses: sp/s per °/s head (*top*), sp/s per °/s retinal slip (*bottom*). (**C**) Schematic of idealized (*black*)

*Figure 4 continued on next page*

*Figure 4 continued*
and "biological" (*blue*) linear temporal filters and their corresponding step responses. Acceleration-like filters perform differentiation, velocity-like filters only change gain, and position-like filters perform leaky integration.

from *Ramachandran and Lisberger, 2005*; *Figure 6A*) as well as to learned changes in Purkinje cell activity during the VOR at low frequencies (data from *Lisberger et al., 1994b*; *Watanabe, 1985*). We note that we were limited to analyzing only low frequency neural data, since neural data for the full range of stimulus conditions analyzed before learning does not exist for after learning. We then compared the learning-related changes in filter shapes across models to assess how the strength of circuit feedback influences the inferred sites and directions of plasticity.

All models, regardless of the strength of efference copy feedback, successfully reproduced behavioral changes in the VOR after learning (*Figure 6A and B*, *Figure 6—figure supplement 1*). Each model captured the learned changes in amplitude and phase of the VOR observed in monkeys across stimulus frequencies (*Ramachandran and Lisberger, 2005*; *Figure 6A and B*). Despite these nearly identical changes in motor output, the underlying circuit plasticity depended critically on the strength of efference copy feedback. Similar to the baseline filters before learning, the net *changes* in the vestibular filters after learning depended on the level of feedback only for pathways in the cerebellar cortex ($k_{PH}$, *Figure 6C*) and not in the brainstem ($k_{EH}$; *Figure 6D*). Most strikingly, the No Feedback model displayed net depression of the vestibular input to Purkinje cells ($k_{PH}$), whereas the Strong Feedback model displayed net potentiation at this site (*Figure 6C*, *right*). Further, there was a gradual transition between the two extremes, with net depression switching to net potentiation at a feedback gain of around 0.4 (*Figure 6C*, *right*). Here, 'depression' and 'potentiation' refer to any combination of changes in excitatory and/or inhibitory inputs that result in decreases and increases, respectively, in the postsynaptic response and do not attempt to disambiguate, for example, between LTD of excitatory inputs and LTP of inhibitory inputs onto the same site (*Carey, 2011*; *Jörntell et al., 2010*). We confirmed that the direction of net plasticity of $k_{PH}$ in each model was necessary by showing that the models failed to reproduce the experimentally observed changes in neural activity during the VOR after learning when the corresponding direction of plasticity was blocked (*Figure 6—figure supplement 3*).

In addition to the net changes in filter strengths, there were also differences in dynamics between the models. The Strong Feedback model displayed *depression* of an acceleration-like component of $k_{PH}$, even though the net change at this filter was potentiation (compare *Figure 6C*, *center* with *Figure 4C*). This observation is consistent with the finding that circuits with strong positive feedback

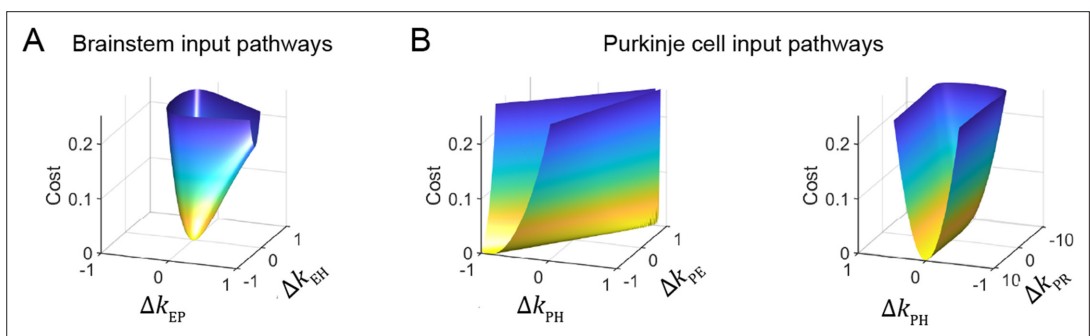

**Figure 5.** Cost function landscape illustrates degeneracy of input pathways to Purkinje cells, but not to the brainstem. Degeneracy in the VOR circuit model fits, illustrated using a simplified model with merged retinal slip and visual prediction pathways (see Materials and methods). Cost is defined as the squared error between model output and experimental data. (**A**) Cost function landscape for the brainstem pathways ($k_{EP}$ and $k_{EH}$) has a single, well-defined minimum defining the best-fit parameter values. $\Delta k_{XY}$ indicates deviations from best-fit values of filter responses at steady state (see Materials and methods). (**B**) Cost function landscape for the Purkinje cell input pathways ($k_{PH}$, $k_{PE}$, and $k_{PR}$) is degenerate, as reflected by the flat valleys of equally well-fit solutions. For each value of the input variables shown ($k_{PH}$ and $k_{PE}$, *left*; $k_{PH}$ and $k_{PR}$, *right*) the value of the third variable ($k_{PR}$ and $k_{PE}$, respectively) was adjusted to minimize the cost. Note that the sharp increase in cost for $\Delta k_{PR}$ occurs where $k_{PR}$ approaches 0, at which point there is no retinal slip input to initiate pursuit behavior.

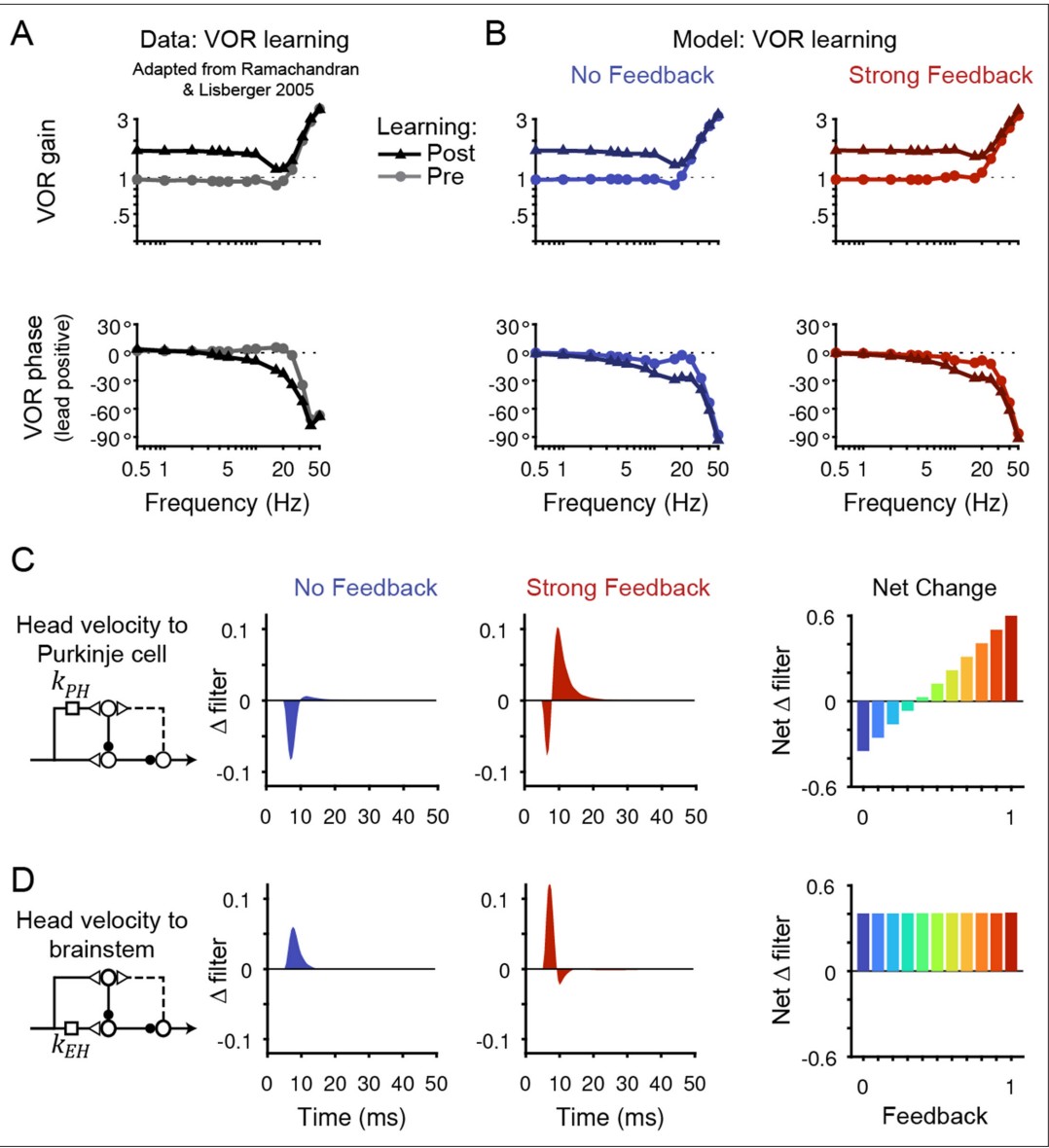

**Figure 6.** Circuit changes underlying learned changes in behavior. (**A,B**) Monkey (**A**) and model (**B**) eye velocity responses to sinusoidal vestibular input before (*gray/light colors, circles*) and after (*black/dark colors, triangles*) VOR learning. Behavior is quantified as the gain and phase of the eye relative to the head (0° phase represents the eye moving exactly opposite to the head). Note that 'VOR gain' is a normalized measure of eye movement amplitude, and is not to be confused with feedback gain. Data are adapted from Figure 4 of ***Ramachandran and Lisberger, 2005***. (**C**) *Left*, Dynamic change in the filter carrying head velocity input to Purkinje cells ($k_{PH}$) after learned increases in the VOR, for the No Feedback (*blue*) and Strong Feedback (*red*) models. Traces represent the *change* in filter strength after learning, rather than the absolute filter shape. *Right*, Net change in the filter $k_{PH}$, calculated by numerically integrating the change in filter shape, for all models. Negative values indicate net depression and positive values indicate net potentiation. (**D**) Same as in (**C**) but for the filter carrying head velocity input to the brainstem ($k_{EH}$). Note that the differences in filter shape between the No Feedback and Strong Feedback models in (**D**) largely reflect high temporal frequencies that were not well constrained by the experiments (see Materials and methods). Changes in filter shapes for intermediate feedback values and for learned decreases in VOR gain are shown in ***Figure 6—figure supplement 2***.

The online version of this article includes the following figure supplement(s) for figure 6:

**Figure supplement 1.** Model response to steps in head velocity after VOR learning.

**Figure supplement 2.** Changes in filter strength after VOR learning for all models.

*Figure 6 continued on next page*

*Figure 6 continued*

**Figure supplement 3.** Changes in neural activity after learning when the direction of plasticity in the cerebellar cortex is restricted.

**Figure supplement 4.** Changes in Purkinje cell activity during the VOR in the dark.

can affect steady-state changes in circuit output through changes in dynamics as well as weights (*Lisberger and Sejnowski, 1992*).

The switch from net depression to net potentiation at $k_{PH}$ with increasing feedback strength arises naturally from the model equations. For a given strength of positive feedback $k_{PE}$ (see Discussion for predicted effects of allowing $k_{PE}$ to change over learning), the learning-related changes in Purkinje cell firing during the VOR in the dark is:

$$\Delta P_{VOR} = \Delta k_{PH} H + \Delta E_{VOR} k_{PE}$$

For models without feedback ($k_{PE} = 0$), depression of $k_{PH}$ is required to reproduce the experimentally observed decrease in Purkinje cell firing during the VOR in the dark ($\Delta P_{VOR} < 0$ requires $\Delta k_{PH} < 0$; *Figure 6—figure supplement 3*; *Figure 6—figure supplement 4A–C*). For models with feedback ($k_{PE} > 0$), the learned change in eye velocity $\Delta E_{VOR}$ will contribute to $\Delta P_{VOR}$. The sign of this contribution is negative (since the eyes rotate even faster opposite to the head after learning) and increases in size with increasing feedback strength $k_{PE}$. For sufficiently large feedback strength $k_{PE}$, the term $\Delta E_{VOR} k_{PE}$ will be more negative than $\Delta P_{VOR}$; hence, the net change in the vestibular filter $\Delta k_{PH}$ must be positive to produce the observed change in Purkinje cell firing (*Figure 6—figure supplement 4D–E*). Thus, both the No Feedback and Strong Feedback models correctly reproduce the observed decrease in Purkinje cell activity, but they use oppositely directed plasticity in the vestibular pathway to do so.

## New explanation for paradoxical changes in the Purkinje cell response to VOR cancellation after learning

The surprising observation that Purkinje cell activity during VOR cancellation increases after VOR increase learning was previously interpreted as evidence that the vestibular inputs to Purkinje cells ($k_{PH}$) must undergo potentiation, rather than depression (*Lisberger, 1994a*; *Miles and Lisberger, 1981*). A limitation of this hypothesis is its implicit assumption that visual input to the Purkinje cells was negligible during VOR cancellation because retinal slip velocity was close to zero. However, for an animal to 'cancel the VOR', it must use visual input to keep its eyes on the target; therefore, even retinal slip that is small but nonzero must have a significant influence somewhere in the oculomotor circuitry and cannot be discounted.

To determine whether such potentiation of the vestibular input to Purkinje cells is necessary to account for the responses of Purkinje cells during VOR cancellation, we examined the response of each model after learning. Note that the models were fit to reproduce learned changes in Purkinje cell activity only during the VOR in the dark, leaving changes in firing during VOR cancellation as a prediction of the model. In contrast to the previous interpretation, we found that all models — both those with potentiation and those with depression of the vestibular input to Purkinje cells, $k_{PH}$ — successfully replicated this increase in Purkinje cell activity (*Figure 7*).

In the Strong Feedback model, the learned increase in Purkinje cell activity during VOR cancellation is not surprising: the vestibular input to Purkinje cells ($k_{PH}$) potentiates, directly driving the increase in firing rate (*Figure 7E–H*). Because eye velocity was small during VOR cancellation and, consistent with experiments, did not change substantially following learning (*Guo and Raymond, 2010*; Materials and methods), efference copy feedback to Purkinje cells made essentially no contribution to the increase in Purkinje cell activity after learning.

More surprisingly, the No Feedback model could also reproduce the observed increase in Purkinje cell activity during VOR cancellation after learning (*Figure 7A and B*), despite net *depression* of the vestibular input to Purkinje cells ($k_{PH}$) (*Figure 6*, *Figure 7C*). This was accomplished by the second feedback pathway in the circuit: external negative feedback of visual error, which drives corrective motor commands (*Figure 7C and D*). The visual feedback loop allowed the VOR to be nearly completely canceled, despite plasticity in the brainstem ($k_{EH}$) and cerebellar cortex ($k_{PH}$) that would otherwise drive

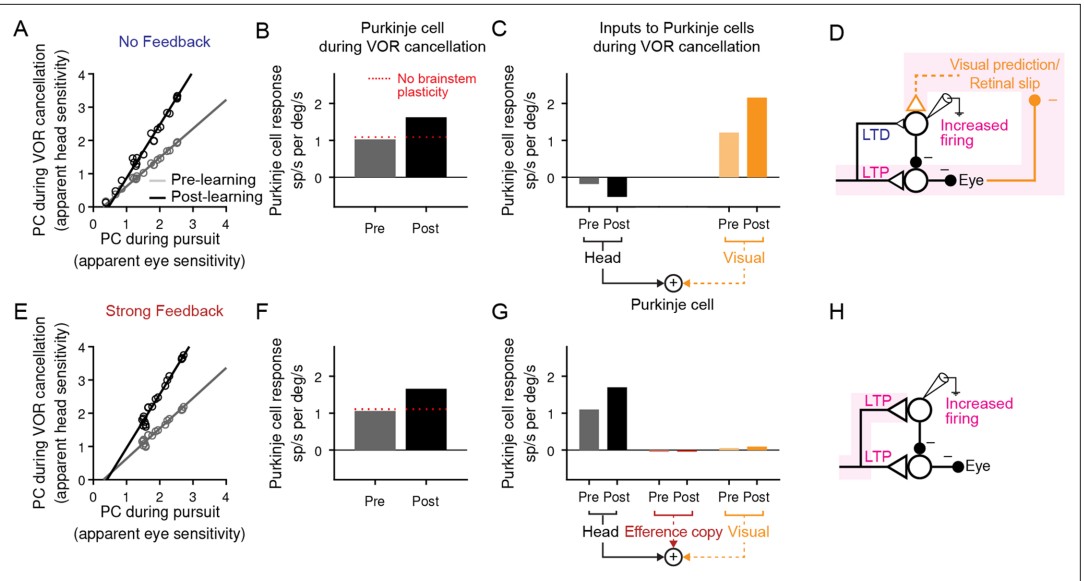

**Figure 7.** Explanation for changes in neural activity during VOR cancellation after learning. (**A**) Response of a population of simulated Purkinje cells during VOR cancellation compared to response during visual pursuit before (Pre) and after (Post) VOR learning for the No Feedback model. Values shown are in units of sp/s per °/s stimulus amplitude. Compare to Figure 13 of *Lisberger et al., 1994b*. (**B**) Average response of Purkinje cell population in (**A**) during VOR cancellation. *Red dashed lines*, model results with brainstem plasticity blocked. (**C**) Inputs to Purkinje cells during VOR cancellation. The visual contribution represents the sum of both the retinal slip and visual prediction pathways. For the No Feedback model, the increase in input through the visual pathway after learning overshadows the decrease in input due to depression in the head velocity pathway. (**D**) Schematic illustrating that in the No Feedback model, the observed increase in Purkinje cell activity is due to negative feedback from vision. (**E, F, G**) Same as (**A, B, C**) for the Strong Feedback model. (**H**) In the Strong Feedback model, the observed increase in Purkinje cell activity is due to potentiation of vestibular inputs to Purkinje cells. Visual and efference copy pathways not shown because their contributions to Purkinje cell activity was minimal.

The online version of this article includes the following figure supplement(s) for figure 7:

**Figure supplement 1.** Explanation for changes in Purkinje cell activity during VOR cancellation in a model without a visual prediction pathway.

larger-than-normal eye movement commands in response to vestibular input. In essence, the visual feedback loop 'works harder' by increasing its input to Purkinje cells after learning. This results in more suppression of the brainstem, thereby counteracting the learned increase in the VOR.

Note that the visual pathways in our model do not undergo plasticity (but see Discussion for implications of considering plasticity in the visual pathways). Thus, consistent with previous experiments (*Lisberger, 1994a*), we do not predict any change in visual pursuit behavior after learning. How do the visual pathways 'work harder' if they do not undergo plasticity? In the simulations shown in *Figure 7*, the visual prediction pathway adjusts its contribution to the eye movement command on the fly in order to maintain cancellation performance. While inclusion of the visual prediction pathways provided a closer quantitative match to the full dataset (see Materials and methods), the qualitative results did not depend upon visual prediction. Specifically, in separate simulations that did not contain a prediction pathway and thus only relied on retinal slip, Purkinje cell activity during VOR cancellation also increased after learning in all models. In this case, Purkinje cell activity increased because VOR cancellation performance became slightly worse over learning, leading to more retinal slip, and thus increased retinal slip feedback to Purkinje cells (*Figure 7—figure supplement 1*). Taken together, this analysis demonstrates that visual negative feedback, regardless of the exact implementation, can cause Purkinje cell activity during VOR cancellation to change in the opposite direction to that expected from the underlying plasticity of vestibular inputs.

In both the No Feedback and Strong Feedback models, the learned increases in Purkinje cell activity during VOR cancellation depended on the existence of plasticity in the direct pathway through the brainstem. When the vestibular input to the brainstem ($k_{EH}$) was held fixed during learning, motor

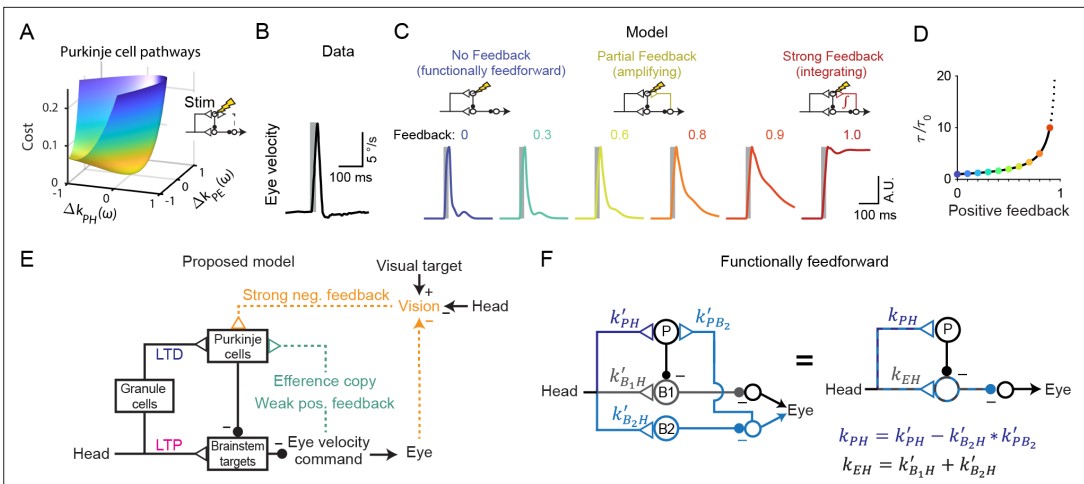

**Figure 8.** Circuit perturbations suggest functionally weak efference copy feedback. (**A**) Cost function landscape for the simplified model when a Purkinje cell stimulation condition is added. Compare to *Figure 5B*. (**B**) Monkey eye velocity in response to pulses of electrical stimulation in the Purkinje cell layer of the floccular complex. Data are adapted from Figure 1B of *Lisberger, 1994a*. (**C**) Model eye velocity responses to brief (25 ms) Purkinje cell stimulation. (**D**) Increase in system time constant due to positive feedback, relative to the time constant without any positive feedback ($\tau_0$). (**E**) Proposed model reconciling previous experimental results. Efference copy feedback is relatively weak, and both net depression in the cerebellar cortex and net potentiation in the brainstem pathway can contribute to learned increases in the VOR. In combination with brainstem plasticity, visual feedback drives the paradoxical increase in Purkinje cell activity during VOR cancellation after learning. (**F**) Feedforward architecture can exist despite anatomical feedback pathways between brain regions. *Left*, a circuit that is functionally feedforward despite anatomical projections that convey an efference copy from a population of brainstem neurons (**B2**) to the Purkinje cells (**P**). This circuit is purely feedforward, since the population of brainstem neurons that receive input from Purkinje cells (**B1**) do not loop back to the cerebellar cortex. *Right*, the equivalent feedforward circuit.

The online version of this article includes the following figure supplement(s) for figure 8:

**Figure supplement 1.** Sensitivity of models to parameter changes.

---

learning was still successful at the level of the eye movement output during the VOR in the dark, but the learned increase in Purkinje cell firing during VOR cancellation no longer occurred (*Figure 7B and F*, dashed lines). This is because eye velocity responses during VOR cancellation do not change substantially after learning (*Guo and Raymond, 2010*). Thus, when brainstem plasticity is intact, Purkinje cell firing must increase to offset the potentiated brainstem pathway. However, if brainstem plasticity is removed, Purkinje cell firing must not change if eye velocity responses are to remain unchanged. Therefore, in each model, two sites of plasticity—in the brainstem and in the cerebellar cortex—were required to explain the apparent paradox that learning decreases Purkinje cell firing measured during the VOR in the dark but increases firing measured during VOR cancellation. Note, however, that while both sites of plasticity are required to fit the current dataset, they are not both required for VOR learning in general (see Discussion, 'Implications for systems consolidation of cerebellar learning'). Overall, this analysis shows that weak feedback models, which predict depression in the vestibular pathway as observed experimentally, are also compatible with the counterintuitive changes in neural activity during VOR cancellation.

## Transient perturbation of activity distinguishes weak and strong feedback

The above results demonstrate that circuit feedback and circuit plasticity are interdependent — knowledge of one constrains the other. Here we use an additional, orthogonal approach to functionally assess circuit feedback by directly perturbing neural activity (*Aksay et al., 2007*; *Sadeh and Clopath, 2020*; *Tsodyks et al., 1997*). A purely feedforward system should only respond briefly to transient stimulation, with a time course dominated by the intrinsic time constants of its neurons and synapses. In contrast, a system dominated by strong positive feedback should instead integrate transient

stimulation, resulting in a prolonged response (*Robinson, 1989*). Mathematically, the response to such a perturbation of activity represents an independent constraint in addition to those provided by sensory-driven changes in activity. As a result, sensitivity analysis demonstrates that the previously observed degeneracy in the strengths of Purkinje cell inputs (*Figure 5C*) is eliminated when a 'Purkinje cell stimulation' condition is added (*Figure 8A*). To leverage this constraint and quantitatively probe the strength of circuit feedback, model simulations were compared to the results of a previous study in which eye movements were evoked by electrical stimulation of floccular Purkinje cells in monkeys (*Lisberger, 1994a*). In this experiment, brief stimulus trains (5 pulses at 200 Hz) applied in the absence of visual or vestibular stimulation evoked smooth eye velocity responses that decayed rapidly after stimulus offset (*Figure 8B*). We mimicked this experiment in our models by increasing Purkinje cell firing rate for 25 ms in the absence of vestibular or visual input. Models with weak or intermediate feedback produced brief eye velocity responses, akin to the data, whereas models with strong feedback produced prolonged eye velocity responses, unlike the data (*Figure 8C*). Although the precise strength of the efference copy feedback is difficult to ascertain because the time constant of decay of responses changes only gradually until the gain of the feedback loop approaches one (*Figure 8D*), this suggests that feedback in this circuit is relatively weak (*Figure 8E*; see Discussion).

## Discussion

Here, we combine a systematic modeling approach with a wide array of neural and behavioral data to demonstrate how changes in neural activity and behavior arise from distributed sites of plasticity in a closed-loop circuit. We find that the presence of internal (*Lisberger, 1994a*; *Miles and Lisberger, 1981*) and external feedback loops can lead to counterintuitive changes in neural activity that mask the underlying sign of plasticity at a given site within a circuit.

Our results reconcile two core models of cerebellar circuit function and learning. Consistent with the Marr-Albus-Ito model and many experimental studies of plasticity (*Coesmans et al., 2004*; *Gilbert and Thach, 1977*; *Ito and Kano, 1982*; *Kimpo et al., 2014*; *Medina and Lisberger, 2008*; *Sakurai, 1987*; *Silva et al., 2023*; *Yang and Lisberger, 2013*; *Yang and Lisberger, 2014*), we find that net depression at the parallel fiber-Purkinje cell synapses can explain all recording and stimulation data, if efference copy feedback is relatively weak or absent. In particular, our model shows how such net depression can be consistent with the paradoxical *increase* in Purkinje cell activity during VOR cancellation, which was previously interpreted as evidence for potentiation of vestibular inputs to Purkinje cells (*Lisberger, 1994a*; *Miles and Lisberger, 1981*). We show how this increase in activity can instead be explained by visual feedback. However, our results also support the presence of a site of plasticity in the brainstem pathway, as proposed in the Miles-Lisberger model. We thus suggest a resolution of a longstanding debate about how the cerebellum implements motor learning, and exemplify more broadly how feedback can obscure underlying plasticity in even a relatively simple closed-loop system.

### Compatibility of weak and strong efference copy feedback models with error-driven parallel fiber-Purkinje cell plasticity

A primary motivation for this study was the question of whether we could identify models in which the inferred changes over learning were consistent with error-driven plasticity in the cerebellar cortex weakening synaptic inputs that are correlated with error. In the classic model of cerebellar learning, climbing fiber activity encoding motor errors drives LTD of parallel fiber inputs to Purkinje cells that were recently active and hence could have contributed to the error (*Ito, 1982*; *Raymond and Medina, 2018*). In this study, we found that only models with weak or no efference copy feedback were compatible with depression of parallel fiber inputs to Purkinje cells, and thus with error-driven plasticity in the cerebellar cortex (see discussion below about alternate implementations of error-driven plasticity that are functionally equivalent). Our study therefore reconciles results from VOR learning with other cerebellum-dependent motor learning paradigms, such as eye blink and smooth pursuit learning, where climbing fiber activity has been less controversially associated with depression of active parallel fiber inputs (*Lisberger, 2021*; *Raymond et al., 1996*; *Raymond and Medina, 2018*).

In contrast, models with strong efference copy feedback predict net potentiation rather than depression of vestibular input to the cerebellar cortex, and make additional predictions that are difficult to reconcile with biology. First, the response to transient stimulation of models with relatively

strong feedback was incompatible with the experimentally observed response (*Figure 8*). Second, the Strong Feedback model with feedback gain equal to 1 is on the brink of instability and thus requires that any change in the strength of the vestibular input to the Purkinje cells ($k_{PH}$) must be precisely offset by a change in the strength of the vestibular inputs to the brainstem ($k_{EH}$) to avoid runaway activity (*Lisberger and Sejnowski, 1992*; *Figure 8—figure supplement 1*).

Despite this evidence against models with strong efference copy feedback, there is almost certainly some efference copy input to the cerebellum. Signals related to motor output have been identified as inputs to the cerebellar cortex (*Giovannucci et al., 2017*; *Person, 2019*). Anatomically, the nucleus prepositus hypoglossi and the medial vestibular nucleus provide eye movement-related mossy fiber afferents to the cerebellar flocculus (*Escudero et al., 1996b*; *Escudero et al., 1996a*). These pathways could reflect true loops, or they could involve 'spiraling' connections between the cerebellum and other regions rather than closed feedback loops onto the same cells, in which case they would be functionally identical to the feedforward model. For example, one population of cells in the medial vestibular nucleus could receive input from Purkinje cells, while a separate population sends output to the cerebellar cortex (*Figure 8F*). To the extent that true feedback loops are present, our analysis suggests that their functional feedback strength is weak (*Figure 6C* and *Figure 8B and C*).

Overall, the present results increase our understanding of VOR learning by predicting the range of efference copy feedback strengths that are compatible with error-driven plasticity at the parallel fiber-Purkinje cell synapse. More generally, a similar strategy could be applied to other learning paradigms in order to determine the quantitative relationship between circuit feedback and plasticity of specific input pathways.

## Implications for systems consolidation of cerebellar learning

Studies of cerebellar learning suggest that memories rapidly form in the cerebellar cortex before being gradually transferred to the brainstem or cerebellar nuclei (*Kassardjian et al., 2005*; *Shutoh et al., 2006*; reviewed in *De Zeeuw et al., 2021*). Such consolidation from a faster-learning site to a slower-learning site is known as systems consolidation and has been shown theoretically to mitigate the 'stability-plasticity dilemma' of allowing fast learning without over-writing old memories. The model presented here represents a single time point of learning, with most of the post-learning experimental data coming from animals that were trained on previous days as well as the day of recording (Materials and methods). Due to this previous training, some consolidation of learning would be expected, consistent with the brainstem plasticity that we inferred. However, a different distribution of plasticity between the cerebellar and brainstem sites would be expected at other times. Before consolidation, learning-related decreases in Purkinje cell firing during the VOR may be supported solely by plasticity in the cerebellar cortex. If this is the case, then our model predicts that 'paradoxical' increases in Purkinje cell responses occurring during VOR cancellation (*Figure 7*) would not be observed at the earliest stages of learning before brainstem plasticity has occurred. Following consolidation, if learning becomes fully independent of the cerebellar cortex, then this has interesting implications for the final sign of plasticity of the vestibular input to Purkinje cells ($k_{PH}$): for the cerebellar cortex to return to its original, pre-learning output, either (1) if there is no efference copy feedback, the vestibular pathway onto Purkinje cells needs to reset to its original strength, resulting in zero net plasticity, or (2) if there is efference copy feedback, the vestibular and efference copy pathways need to provide exactly canceling inputs during performance of the VOR. In the latter case, in the absence of plasticity in the efference copy pathway, potentiation in the vestibular pathway ($k_{PH}$) would be required to offset changes in input from the efference copy pathway due to brainstem plasticity. This provides the possibility that the net sign of plasticity in vestibular inputs to Purkinje cells could be depression at early stages of learning and potentiation after complete consolidation of learning.

## Relation to other sloppy models

In complex biological circuits, many different configurations of biological parameters can enable a circuit to accomplish the same task. This results in degeneracy in the relation between model parameters and circuit output, known as 'sloppiness in model fits' (*Bittner et al., 2021*; *Costa et al., 2013*; *Fisher et al., 2013*; *Foster et al., 1993*; *Goldman et al., 2001*; *Gonçalves et al., 2020*; *O'Leary and Marder, 2016*; *Prinz et al., 2004*). Here we take two approaches to the issue of sloppiness: we systematically fix a key model parameter—the strength of efference copy feedback—to eliminate

degeneracy, and also study a simpler analytic model for which we could fully derive the cost function landscape (*Figure 5*). We show that the presence of multiple feedforward and feedback pathways converging on the same site can lead to sloppiness in inferring the sites and signs of plasticity, but that additional data from perturbations may constrain the most fundamental aspects of this sloppiness.

Such degeneracy may also be present in previous work that used multiple linear regression analysis to assess the strength of efference copy feedback. In such analyses, Purkinje cell activity was fit to linear combinations of position, velocity, and/or acceleration of vestibular input, retinal slip, and eye movement ('efference copy input'; *Blazquez et al., 2003*; *Hirata and Highstein, 2001*). These analyses found non-zero values for the coefficients representing eye velocity, which has been interpreted as evidence for efference copy feedback to Purkinje cells. Such studies used only a single frequency sinusoidal stimulus, making position and negative acceleration predictors strongly correlated, and delays and phase shifts difficult to distinguish. More fundamentally, these models are a special case of the more general linear filter models analyzed here, for which there is an inherent non-identifiability of the inputs to Purkinje cells.

## Model assumptions

### Pathways undergoing plasticity

Our model only included plasticity in the pathways carrying vestibular information ($k_{PH}$ and $k_{EH}$), which have been the focus of debate about VOR learning. However, it is reasonable to expect that the efference copy and visual inputs to Purkinje cells may be governed by the same correlational plasticity rule thought to control plasticity at the vestibular parallel fiber-Purkinje cell synapses, in which elevated climbing fiber activity drives depression of recently active parallel fiber inputs to Purkinje cells. We did not include plasticity of these inputs because they were not needed to fit the available data. However, we can make qualitative predictions about the effects of applying the correlational plasticity rule to these inputs.

We first consider the correlational plasticity rule applied to the efference copy pathway. This pathway ($k_{PE}$) is driven by ipsiversive eye velocity, whereas climbing fiber activity in the flocculus is driven by contraversive retinal slip and suppressed by ipsiversive slip (*Raymond and Lisberger, 1998*). During learning to increase the VOR, ipsiversive eye movements are accompanied by ipsiversive retinal slip. Hence, activation of parallel fibers carrying efference copy signals should be accompanied by little or no climbing fiber activity, which *in vitro* has been shown to induce potentiation of the parallel fibers inputs to Purkinje cells (*Crepel and Jaillard, 1991*; *Suvrathan et al., 2016*). Thus, we predict net potentiation of $k_{PE}$ for learned increases in the VOR. Applying similar logic to learned decreases in the VOR predicts net depression of $k_{PE}$. However, since the amplitude of eye movements is small during training to decrease the VOR, the plastic changes may be smaller in this case. In both cases, the changes in $k_{PE}$ are in the correct direction to support learning of amplified or attenuated eye movements, respectively. Changes in $k_{PE}$ would also lead to alterations of VOR dynamics, as well as amplitude, due to changing the strength of the efference copy feedback loop (*Lisberger and Sejnowski, 1992*).

For the visual pathway, the correlational learning rule predicts potentiation of $k_{PR}$ during both learned increases and decreases in the VOR, because visual error simultaneously drives climbing fiber and parallel fiber input in the same direction, so parallel fiber-climbing fiber correlations are independent of the sign of visual error. This may explain the increase in Purkinje cell firing during smooth pursuit that is observed after *both* directions of VOR learning (*Blazquez et al., 2003*; *Lisberger et al., 1994b*; *Miles et al., 1980*).

We note that the correlational learning rule discussed here involves another type of feedback in the VOR circuit: motor errors are fed back to the cerebellar cortex from the external world via climbing fibers. Such feedback for learning operates on a slower timescale than the feedback of the efference copy and visual pathways in our model, which affect circuit dynamics on the fast timescale of behavior. Thus, our conclusions that efference copy feedback is likely not strong in this circuit does not preclude an important role for other types of feedback, including feedback for learning (*Boven et al., 2023*).

### Implementation of error-driven plasticity in cerebellar cortex

Our model is agnostic to the biological mechanism by which the vestibular inputs to Purkinje cells undergo net 'depression'. For example, instead of the parallel fiber-Purkinje cell synapse undergoing

LTD, the parallel fiber-interneuron synapse or the interneuron-Purkinje cell synapse could undergo LTP (*Carey, 2011*; *Jörntell et al., 2010*). Additionally, instead of ipsiversive vestibular inputs undergoing LTD, contraversive vestibular inputs could undergo LTP, since there are two cerebellar flocculi located bilaterally which converge to move the eyes. Therefore, our models with weak efference copy feedback are consistent with multiple plasticity mechanisms, including a contribution to VOR learning from LTP (*Boyden et al., 2004*; *De Zeeuw, 2021*; *du Lac et al., 1995*; *Titley et al., 2010*).

### Visual input

We modeled visual input only to the cerebellar cortex, motivated by studies that show large decreases in the amplitude of smooth pursuit eye movements following lesions of the floccular complex (*Belton and McCrea, 2000*; *Burde et al., 1975*; *Estanol et al., 1979*; *Rambold et al., 2002*; *Westheimer and Blair, 1974*; *Zee et al., 1981*). There is some anatomical evidence for direct visual input to the brainstem as well (*Balaban, 1983*), and our model could be extended to implement this. This additional visual pathway would make the inputs to the brainstem site degenerate when fitting to neural and behavioral data alone, similar to the degeneracy demonstrated among the three pathways onto the Purkinje cells. Models fit to data collected after complete lesions of the floccular complex, or synaptic blockade of synaptic input to Purkinje cells, could provide an additional constraint to resolve this degeneracy.

## Experiments to further test the strength of efference copy feedback

Our two lines of evidence for weak efference copy feedback—compatibility with climbing fiber driven LTD in the cerebellar cortex, and brief responses to transient stimulation—are both consistent with a range of weak to moderate efference copy feedback strengths. The functional impact of such feedback to Purkinje cells could be more directly assessed by comparing the motor response to Purkinje cell stimulation in the presence versus absence of parallel fiber input. One study in mice did just that by blocking granule cell activity, which appeared to have little effect on the dynamics of the motor response to Purkinje cell stimulation, supportive of a weak feedback model in mice, although this was not quantified (*Wada et al., 2014*). If efference copy pathways can be anatomically identified, a direct experimental test of strong versus weak feedback would be to measure the impact of eliminating those inputs. Finally, as described above, our weak feedback models predict that if VOR cancellation is measured immediately after training, before consolidation has occurred, then paradoxical increases in Purkinje cell activity will no longer be observed.

## Materials and methods

### Data set

Four datasets were used to capture neural and behavioral dynamics before and after learning. First, neural and behavioral data before learning ('Dataset 1'; shown in *Figure 3*) were obtained from two male rhesus monkeys (*Macaca mulatta*) trained to perform a visual fixation task. A subset of this dataset has been published previously (*Kimpo et al., 2014*; *Raymond and Lisberger, 1998*). Briefly, neural responses were recorded extracellularly from Purkinje cells in the floccular complex of the cerebellar cortex while the monkeys made horizontal eye movements in response to various combinations of visual and vestibular stimuli. Vestibular stimuli consisted of passive whole-body rotation in the horizontal plane. Visual stimuli consisted of a horizontally moving target subtending 0.5° of visual angle, which was accompanied by a larger black-and-white pattern subtending 20° to 30° of visual angle for all stimulus conditions except for smooth pursuit. Four combinations of visual and vestibular stimuli were delivered: head movements in the dark ('Vestibular only', which elicits the VOR), visual target motion with the head stationary ('Visual only', which elicits smooth pursuit eye movements), visual target and head motion at the same speed but in opposite directions (*Figure 3*, 'Vestibular + visual', also referred to as ×2 because accurate tracking requires compensatory eye movements that are twice as large as normal), and visual target and head motion at the same speed and in the same direction (*Figure 3*, 'Vestibular – visual', also referred to as ×0 because accurate tracking requires *no* rotation of the eyes in their sockets, and also known as 'VOR cancellation' since normal VOR eye movements are suppressed). These combinations were delivered as sine waves in stimulus velocity with frequencies of 0.5 Hz, 2 Hz, 5 Hz, and 10 Hz at ±10 °/s, or as steps in stimulus

velocity with durations of 80 ms, 150 ms, 250 ms, and 500 ms at 15 °/s. Smooth pursuit data were only available for 0.5 Hz sine waves (delivered at ±31.4 °/s), resulting in a total of 25 distinct conditions. Eye position (angle of the eye relative to the head) was measured using the scleral search coil method.

Eye velocity and neural activity from Dataset 1 were further processed as follows. Eye velocity was calculated by differentiating eye position. Saccades were removed from eye velocity traces using an automatic threshold algorithm with a threshold of 30 °/s. To allow comparison across datasets and with previous studies, we analyzed horizontal gaze velocity Purkinje cells (HGVPs), which are the largest subpopulation of floccular Purkinje cells and are widely considered to be important for horizontal gaze control (*Katoh et al., 2015*; *Lisberger et al., 1994b*). Purkinje cells were considered HGVPs if (1) during smooth pursuit, firing rate was modulated by at least ± 10 sp/s and the phase difference between peak firing rate and peak ipsiversive eye velocity was less than 45°; (2) during VOR cancellation, firing rate was modulated by at least ± 10 sp/s and the phase difference between peak firing rate and peak ipsiversive head velocity was less than 45°; and (3) firing rate modulation was greater during horizontal than during vertical smooth pursuit. Purkinje cell firing rates were calculated by convolving raw simple spikes times with a 10 ms standard deviation Gaussian filter. Baseline firing rates were removed by subtracting a moving average calculated over an 11 s window. Eye velocity and Purkinje cell firing rates were then averaged across stimulus cycles for each cell. Neurons were only recorded on one side of the brain, so to account for the corresponding population in the opposite hemisphere, Purkinje cell responses to ipsiversive stimulation were averaged together with the inverted response to contraversive stimulation. Finally, data were averaged across all cells to create mean Purkinje cell firing rate and mean eye velocity traces for each stimulus condition.

The remaining datasets were used to fit changes that occurred after learning. Dataset 2 was taken from *Ramachandran and Lisberger, 2005* and consisted of VOR *behavior only* in *Macaca mulatta* before and after learning. In that study, the VOR was tested at 15 different frequencies from 0.5 to 50 Hz in three monkeys, both before learning and after several days of training to increase or decrease the gain of the VOR. We averaged the reported gain and phase of the eye movement response across all three monkeys (*Figure 6A*). Note that 'VOR gain' is a normalized measure of eye movement amplitude during the VOR, not to be confused with efference copy feedback gain. The 12.5 Hz point was excluded as an outlier since it was reported to likely be influenced by a mechanical issue and it was not seen in the one monkey tested using a more reliable vestibular stimulation technique (*Ramachandran and Lisberger, 2005*).

Datasets 3 and 4 consisted of changes in *Purkinje cell activity* after learning from *Lisberger et al., 1994b* and *Watanabe, 1985*. Both of these studies characterized the amplitude of Purkinje cell activity during the VOR before and after learning in monkeys (*Macaca mulatta* and *Macaca fuscata*, respectively), using low frequency 0.3 Hz sine waves (*Watanabe, 1985*) or steps (*Lisberger et al., 1994b*) in vestibular stimulus velocity. Both the steady-state changes (averaged 100–200 ms after step onset) and transient changes (peak firing rate measured 0–50 ms after step onset) after learning reported by *Lisberger et al., 1994b* were included in the cost function. Learning-related changes in neural activity at higher sinusoidal frequencies have not been reported. To account for different amounts of behavioral learning (change in VOR gain) in different studies, we normalized the change in Purkinje cell modulation by the change in VOR performance. The resulting experimentally reported changes were: for the 0.3 Hz sine waves experiments, 0.72 sp/s per °/s head velocity per fractional change in VOR gain for VOR-decrease learning and −0.55 for VOR-increase learning during 0.3 Hz sine waves (*Watanabe, 1985*); and for the step response experiments, 1.36 for VOR-decrease learning and −0.75 for VOR-increase learning for the sustained component, and 1.62 for VOR-increase learning and −1.01 for VOR-decrease learning for the transient component (*Lisberger et al., 1994b*). The low-frequency results from the two studies were averaged to yield target changes of 1.04 sp/s per °/s head velocity per fractional change in VOR gain during VOR-decrease learning, and −0.65 during VOR-increase learning. A large penalty was applied for deviations from the desired change in steady state Purkinje cell modulation. This led to the desired changes being achieved after learning (1.04 sp/s per °/s and −0.65 sp/s per °/s during VOR-decrease and VOR-increase learning, respectively). A change in behavior of 60% (e.g. VOR gain increase from 1.0 to 1.6) was simulated, comparable to the changes achieved in these studies.

## Model implementation

### Model structure

The model architecture was constrained by anatomy, similar to previous models (*Clopath et al., 2014*; *Dean and Porrill, 2008*; *Lisberger, 1994a*; *Lisberger and Sejnowski, 1992*; *Miles and Lisberger, 1981*; *Tabata et al., 2002*; *Yamazaki and Nagao, 2012*; *Figure 2*). Anatomically, in the direct pathway, vestibular signals (*H*) from the semicircular canals travel through Scarpa's ganglion to excite the ipsilateral vestibular nuclei, which directly inhibit motor neurons in the oculomotor nuclei to produce contraversive eye movements (*E*). The motor output from this direct pathway is modified by the indirect pathway through the cerebellum, in which the vestibular signals travel through the cerebellar cortex via granule cells and Purkinje cells, before reuniting in the vestibular nucleus. In addition to vestibular signals, Purkinje cells in the floccular complex of the cerebellar cortex also receive visual and eye velocity-related signals as described in the main text.

Note that Purkinje cells have an unusually high baseline firing rate (~50–100 Hz); in our model, this baseline is subtracted, and bidirectional modulation of firing rate is represented by values above and below zero. In the biological circuit, both Purkinje cells and their target neurons in the vestibular nuclei are tonically active, due to intrinsic excitability (*Bagnall et al., 2008*). Thus, decreases in Purkinje cell firing release the target neuron from inhibition, causing the target neuron to fire more. Similarly, in the circuit model, the relevant variable is the change in Purkinje cell firing relative to baseline, rather than the absolute firing rate, and the model Purkinje cells are still effectively inhibitory.

Based on evidence for linearity in the VOR circuit and in cerebellar Purkinje cells (*Bagnall et al., 2008*; *du Lac and Lisberger, 1995*; *Lisberger and Fuchs, 1978*; *McElvain et al., 2015*; *Payne et al., 2019*; *Walter and Khodakhah, 2006*), the model architecture is described by the following linear firing rate equations:

$$\hat{E}(t) = -(H * k_{EH})(t) + (P * k_{EP})(t) \tag{1}$$

$$\hat{P}(t) = (H * k_{PH})(t) + (R * k_{PR})(t) + (T * k_{PT})(t) + (E * k_{PE})(t) \tag{2}$$

where * indicates temporal convolution, *E* is eye velocity, *H* is head velocity, *P* is Purkinje cell simple spike rate, *R* is retinal slip velocity, *T* is visual target velocity, and $k_{XY}$ indicates the linear temporal filter to *X* from *Y*. $\hat{E}(t)$ and $\hat{P}(t)$ represent the model predictions for eye velocity and Purkinje cell firing rate. On the right side of the equation, as discussed in more detail below, the initial fits to the model used the recorded eye velocity *E* and Purkinje cell activity *P*. These initial fits were then replaced by a full closed-loop model in which both the left and right sides of the equations self-consistently used the model eye velocity and Purkinje cell activity. See the Transient Stimulation section below for modified equations incorporating the Purkinje cell stimulation condition.

The temporal filters $k_{EH}$, $k_{EP}$, $k_{PH}$, and $k_{PR}$ were parameterized using a raised cosine basis (*Pillow et al., 2005*) to efficiently capture biological filter shapes with relatively few parameters (10 basis vectors per filter for $k_{PH}$, $k_{EH}$, and $k_{EP}$; 12 basis vectors for $k_{PR}$). Each basis vector was given by:

$$B_i(t) = \begin{cases} \dfrac{\cos(log[t + \psi] - \phi_i) + 1}{2} & \text{for t such that } log(t + \psi) \in [\phi_i - \pi, \phi_i + \pi] \\ 0 & \text{otherwise} \end{cases}$$

Each filter was then constructed by summing the basis vectors multiplied by their corresponding weights $b_i$:

$$k_{XY}(t) = \sum_i b_i B_i(t)$$

The time axis was logarithmically spaced so that the temporal filters were represented in greater detail at short timescales. The area of each basis vector was normalized to equal one. The duration of each filter was limited to $t_{max}$ = 50 ms for $k_{PH}$ and $k_{EH}$, 150 ms for $k_{EP}$, and 500 ms for $k_{PR}$. These durations were chosen on the basis of preliminary model fits using longer durations, which nonetheless resulted in shorter optimal filters. The absolute minimum latency of each temporal filter was set to 5 ms for $k_{PH}$ and $k_{EH}$, and 1 ms for $k_{EP}$, which resulted in similar latencies to the peak of the first basis vector of 7 ms for $k_{PH}$ and $k_{EH}$ and 7.5 ms for $k_{EP}$, with latencies to the half-max of the first basis vector

of 6 ms for $k_{PH}$ and $k_{EH}$ and 4 ms for $k_{EP}$, generally consistent with reported latencies in the VOR circuit (*du Lac et al., 1995*; *Lisberger, 1984*). The minimum latency of $k_{PR}$ was set to 60 ms (*Krauzlis and Lisberger, 1994*).

For the temporal filter conveying efference copy feedback to Purkinje cells, $k_{PE}$, the temporal dynamics were represented by an exponential filter with a time constant of 3 ms, approximating a fast monosynaptic connection in order to allow the full range of frequencies of the VOR stimulus to be well-fit by the Strong Feedback model (*Figure 6A and B*). The amplitude of this filter was set to make the total steady-state gain of the efference copy feedback loop vary from zero ('No Feedback') to one ('Strong Feedback') in steps of 0.1 to create a series of otherwise identical models. The total steady-state feedback gain corresponded to the time integral (or, in discretized time, sum) of the convolution of the two filters comprising the feedback loop, $k_{PE}$ and $k_{EP}$: $g = \sum_{t=0}^{Tmax} (k_{EP} * k_{PE})(t)$. For each model, the relative strength of $k_{PE}$ and $k_{EP}$ could vary, but the total steady-state gain $g$ was held fixed. Note that the gain of this feedback loop is positive: the Purkinje cell to brainstem pathway is inhibitory (because Purkinje cells are inhibitory), the brainstem to eye velocity command pathway is inhibitory (to achieve counter-rotation of the eyes in response to head turns), and the feedback of this eye velocity command back to Purkinje cells ($k_{PE}$) is positive. Thus, the loop overall represents positive feedback, as proposed by *Miles and Lisberger, 1981*.

For the visual prediction pathway $k_{PT}$, which represents visual prediction signals, sometimes referred to as 'extraretinal' signals, we parameterized the filter as follows. For the experimental paradigms using sinusoidal visual stimuli, we fit the amplitude and phase of $k_{PT}$ separately for each frequency in the data (0.5 Hz, 2 Hz, 5 Hz, and 10 Hz). For the paradigms using steps in visual stimulus velocity, $k_{PT}$ was a single exponential filter with time constant 25 ms and a delay of 60 ms to account for the unpredictable nature of the step onset, and with amplitude as a free parameter that we fit. We note that the visual prediction filters for the sinusoidal and step stimuli are different from each other and were allowed to have different values before and after learning—this reflects our assumption that the visual prediction mechanism is able to nonlinearly adjust in order to account for different behavioral demands.

Inclusion of the visual prediction pathway, and our assumption that it is flexible, is motivated by several behaviors that cannot be explained by retinal slip alone. First, during smooth pursuit, primates can follow complex yet predictable visual stimuli with small (~10 ms) or even negative (anticipatory) latencies (*Leung and Kettner, 1997*). Second, during smooth pursuit of a visual target that goes behind an occluder, eye velocity can change in anticipation of target reappearance, even anticipating expected changes in target velocity (*Becker and Fuchs, 1985*; *Bennett and Barnes, 2004*). Finally, anticipatory smooth eye movements can even occur in response to audio cues or stationary visual cues that predict upcoming target movement (*Boman and Hotson, 1988*; *Jarrett and Barnes, 2005*). Additionally, there is evidence that such signals exist in the brain: visual prediction signals have been observed at several sites that provide input to the floccular complex: in the medial superior temporal area (MST), a higher order region of motion processing in cortex (*Kawano et al., 1994*; *Newsome et al., 1988*; *Sakata et al., 1983*), in the dorsolateral pontine nucleus (DLPN), a brainstem nucleus that projects to the flocculus (*Mustari et al., 1988*), and at mossy fiber inputs within the flocculus (*Noda, 1986*). The visual prediction pathway was therefore included to explain the behaviors above, and to provide the best quantitative fits to visual stimuli before learning, particularly at higher frequencies. Our main conclusions did not depend on this pathway, however. If the visual prediction pathway is excluded, then the inferred plasticity in the vestibular pathways and the explanation for the paradoxical changes in Purkinje cell activity are both unchanged.

## Model fitting

Model fitting was performed in two steps. First, the filters were initialized either using linear optimization with the model in open-loop configuration (i.e. with separate fits of *Equations 1 and 2*) or using the results of the fit to a model with a similar level of efference copy feedback (i.e. in the Figures, the final fits for the model with g = 0 were used to initialize the fits to the model with g = 0.1, the results from g = 0.1 were used to initialize the fits for g = 0.2, etc.). Note that model fits were similar for both initializations, differing only in high-frequency components that are not well-constrained by the data. Second, the filters were fine-tuned using nonlinear optimization with the model in closed-loop configuration (i.e. with both *Equations 1 and 2* operating simultaneously).

In the open-loop initialization step, we first performed two separate linear regressions corresponding to *Equations 1 and 2* to provide initial estimates of the filters $k_{PH}$, $k_{EH}$, $k_{PR}$, $k_{PT}$, and $k_{EP}$, using only data before learning. (Note that the open loop initialization step requires both eye and neural data for each condition. Since Dataset 1 contains both eye and neural data only for frequencies ≤10 Hz, whereas Dataset 2 contains frequencies up to 50 Hz but only consists of behavioral data, we only included the subset of Dataset 2 ≤10 Hz in this initialization step. The full dataset was used for the final closed loop model fits.) We then performed an additional pair of linear regressions corresponding to *Equations 1 and 2* to provide initial estimates of the changes in the filters $k_{PH}$ and $k_{EH}$ after learning.

To keep the fit coefficients on a similar scale, each signal type used to fit the model (head velocity, target velocity, eye velocity, retinal slip velocity, and Purkinje cell firing rate) was normalized before fitting by dividing by its standard deviation, calculated across all 25 conditions before learning. Because the mean Purkinje cell firing rate was already subtracted, and the velocity signals were centered on zero, mean subtraction was not needed before normalization. The learned filters were converted back to real-world units after the fit was complete.

To discourage overfitting of limited data coming from multiple animals and datasets, regularization was applied using Tikhonov matrices to minimize both the amplitude of the weights and the second derivative of the weights, with the following parameters. The regularization penalty for the first basis vector before learning equaled 1 for $k_{EH}$, $k_{PH}$, and $k_{PT}$; 6 for $k_{PR}$; 40 for $k_{EP}$; and 0.25 for $k_{PT}$. These values were chosen empirically by increasing penalties for $k_{EP}$ and $k_{PR}$ until the resulting filters were reasonably smooth. The resulting range in regularization penalties was likely due to the large differences in the final filter weights (*Figure 4A and B*) and timescales (maximum time ranging from 50 to 500 ms). To encourage filters to be brief, the regularization penalties for $k_{EH}$, $k_{PH}$, and $k_{PR}$ increased with the square of the basis index, so that longer timescale basis vectors were penalized more heavily than shorter timescale basis vectors.

Since neural data was not available for the full range of VOR frequencies after learning, the high-frequency components of the post-learning linear filters were under-constrained. To discourage this flexibility from leading to artifactual differences between filters before versus after learning, we used a separate regularization penalty to encourage filter weights after learning to be similar to those before learning. This regularization penalty was applied to the differences between the before-learning weights and the after-learning weights using the same Tikhonov matrices described above, but with all regularization penalties scaled ×12 to encourage relatively small changes in filter coefficients compared to the baseline values of the filter coefficients.

In the second, closed-loop fitting step, the filter coefficients were initialized as described above and then, as described below, fine-tuned through a nonlinear fitting procedure conducted with the model in closed-loop configuration. The non-visual filters were fine-tuned separately from the visual filters to reduce the number of parameters that needed to be optimized simultaneously.

To fine-tune the non-visual filters, the filters $k_{PH}$, $k_{EH}$, and $k_{EP}$ before learning and the change in the filters $\Delta k_{PH}$ and $\Delta k_{EH}$ after learning were simultaneously fine-tuned in a single step using the MATLAB function *fmincon*, which optimizes nonlinear constrained problems using the Interior Point Algorithm. Only stimulus conditions in the dark were included at this stage. These filters were optimized according to the following cost functions:

$$
\begin{aligned}
C_{\text{PreLearn}} \quad &= \sum_{i=1}^{n} \sum_{t=1}^{T} (\hat{E}_i(t) - E_i(t))^2 + \sum_{i=1}^{n} \sum_{t=1}^{T} (\hat{P}_i(t) - P_i(t)) \\
&+ \sum_{j=1}^{m} \left( ln(\hat{E}_{gain,j}^{pre}) - ln(E_{gain,j}^{pre}) \right)^2 + \sum_{j=1}^{m} \left( \hat{E}_{ph}^{pre} - E_{ph}^{pre} \right)^2 \\
&+ \sum_{j=1}^{m} \left( ln(\hat{P}_{gain,j}^{pre}) - ln(P_{gain,j}^{pre}) \right)^2 + \sum_{j=1}^{m} \left( \hat{P}_{ph}^{pre} - P_{ph}^{pre} \right)^2 \\
&+ \left( \hat{P}_{SS}^{pre} - P_{SS}^{pre} \right)^2 + \left( \hat{E}_{SS}^{pre} - E_{SS}^{pre} \right)^2
\end{aligned}
$$

$$
\begin{aligned}
C_{\text{PostLearn}} \quad &= \sum_{j=1}^{m} \left( \left( ln\left(\hat{E}_{gain,j}^{post}\right) - ln\left(\hat{E}_{gain,j}^{pre}\right) \right) - \left( ln\left(E_{gain,j}^{post}\right) - ln\left(E_{gain,j}^{pre}\right) \right) \right)^2 \\
&+ \sum_{j=1}^{m} \left( \left( \hat{E}_{ph,j}^{post} - \hat{E}_{ph,j}^{pre} \right) - \left( E_{ph,j}^{post} - E_{ph,j}^{pre} \right) \right)^2 \\
&+ \sum_{j=1}^{m} \left( \left( ln\left(\hat{P}_{gain,j}^{post}\right) - ln\left(\hat{P}_{gain,j}^{pre}\right) \right) - \left( ln\left(P_{gain,j}^{post}\right) - ln\left(P_{gain,j}^{pre}\right) \right) \right)^2 \\
&+ \sum_{j=1}^{m} \left( \left( \hat{P}_{ph,j}^{post} - \hat{P}_{ph,j}^{pre} \right) - \left( P_{ph,j}^{post} - P_{ph,j}^{pre} \right) \right)^2 \\
&+ \left( \hat{P}_{trans}^{post} - P_{trans}^{post} \right)^2 + \left( \hat{P}_{SS}^{post} - P_{SS}^{post} \right)^2 + \left( \hat{E}_{SS}^{post} - E_{SS}^{post} \right)^2
\end{aligned}
$$

In $C_{\text{PreLearn}}$, the first two terms correspond to fitting estimates (denoted by ^) to the eye velocity ($E_i$) and Purkinje cell activity ($P_i$) before learning for each of the $n$ stimulus conditions $i$ shown in **Figure 3** (Dataset 1), with squared error summed over all time points $t$ from 1 to $T$ in each stimulus condition. The third and fourth terms correspond to fitting the gain and phase of eye velocity ($E$) during the VOR before learning for each of the $m$ frequencies $j$ in Dataset 2 (**Figure 6A**). The gain and phase were calculated by fitting a sine wave to the model eye velocity or Purkinje cell activity, for example $P = a_1 sin\left(2\pi ft\right) + a_2 cos\left(2\pi ft\right)$. The fifth and sixth terms correspond to enforcing the gain and phase of Purkinje cell activity during the VOR in the dark at 0.5 Hz from Dataset 1. The final two terms enforce the steady state Purkinje cell ($P_{SS}$) activity and eye velocity ($E_{SS}$) during the VOR (extracted from the 500 ms step in head velocity in Dataset 1).

$C_{\text{PostLearn}}$ was calculated for both increases and decreases in VOR gain. The first two terms of $C_{\text{PostLearn}}$ correspond to fitting the *changes* in gain and phase of eye velocity during the VOR over learning for each frequency in Dataset 2. The third and fourth terms correspond to enforcing the *changes* in the gain and phase of Purkinje cell activity during the VOR at 0.5 Hz (Dataset 1 and Datasets 3 and 4). The fifth term encourages the transient Purkinje cell responses to steps ($P_{trans}^{post}$) to mimic the transient responses after learning reported in Dataset 3 (**Lisberger et al., 1994b**). The final two terms enforce the steady state Purkinje cell activity and eye velocity during the VOR after learning (Datasets 3 and 4).

The cost functions above, $C_{\text{PreLearn}}$ and $C_{\text{PostLearn}}$, were summed together in order to calculate the total cost for the simultaneous closed-loop model fitting of the non-visual filters before and after learning during conditions in the dark.

To fine tune the visual filters, the following cost function was optimized for filters $k_{\text{PR}}$ and $k_{\text{PT}}$ before learning using the Nelder-Mead simplex method (MATLAB function *fminsearch*):

$$
C = \sum_{i=1}^{n} \sum_{t=1}^{T} \left( \hat{E}_i\left(t\right) - E_i\left(t\right) \right)^2 + \sum_{i=1}^{n} \sum_{t=1}^{T} \left( \hat{P}_i\left(t\right) - P_i\left(t\right) \right)^2
$$

Here, only stimulus conditions that include visual input (all conditions except VOR in the dark) are included. The visual parameters were fine-tuned in this separate step to allow faster optimization, since increasing the number of parameters that must be tuned in a single optimization step greatly increases the time required. The vestibular-only conditions were enough to constrain the strengths of the non-visual weights, since the strength of positive feedback was fixed in each model, and thus the process could be split into these two steps.

The models were fit to reproduce learned changes in Purkinje cell firing rate over learning only during the VOR in the dark, leaving changes in firing rate during cancellation as a prediction of the model. Previous studies show that the amplitude of eye movements during VOR cancellation at low frequencies (0.5 Hz) is unchanged after learning (**Guo and Raymond, 2010**). To replicate this observation and ensure that the motor performance during VOR cancellation at low frequencies did not change after learning, we assumed that the visual prediction pathway $k_{\text{PT}}$ contributed to maintenance of constant VOR cancellation performance after learning. This was enforced by adjusting $k_{\text{PT}}$ after learning to minimize the cost function:

$$
C = \sum_{t=1}^{T} \left( \hat{E}^{post}\left(t\right) - \hat{E}^{pre}\left(t\right) \right)^2
$$

where $E$ represents the eye velocity during VOR cancellation (×0 condition, see *Figure 3*) for 0.5 Hz sinusoidal stimuli before (pre) or after (post) learning. We emphasize that allowing different values for $k_{PT}$ after learning does not imply that plasticity is required in the visual pathway. Instead, the changes in the visual prediction signal reflect the ability of primates to accurately anticipate the visual tracking command required to follow motion of a repeatable stimulus, even after learned changes in the VOR. For example, after VOR-increase learning, larger visually driven eye movement commands are required to cancel the VOR during VOR cancellation, but smaller visually driven eye movement commands are required to track the '×2' stimulus delivered during training. Conceptually, this switch does not require additional training or plasticity of the visual prediction pathway. Instead, it reflects the ability of the visual system to make online predictions to guide behavior.

Simulations were run by calculating the convolutions in *Equations 1 and 2* numerically with 0.5 ms time steps. In *Figure 3C*, the root mean square error (in units of sp/s or °/s) was normalized by dividing the model error by the maximum stimulus speed (in units of °/s) during each behavioral condition and then averaging across conditions.

## Schematic of idealized and 'biological' linear temporal filters

The schematics illustrating idealized and 'biological linear' temporal filters in *Figure 4C* are for pedagogical purposes, to aid in the interpretation of the model filters and step responses in *Figure 4A and B*. The idealized filters were convolved with a step function to produce the idealized step response. The biological filters for acceleration and velocity were constructed by convolving the idealized filters with an exponential filter, twice. The biological filter for position was constructed manually. All biological filters were then convolved with a step function to produce the 'biological' step response.

## Model analysis

To mathematically illustrate the degeneracy of the VOR circuit, we analyzed a slightly simplified version of the model with no explicit visual prediction pathway or explicit consideration of delays (which would provide a linear delay element), so that the filter components at all complex frequencies $s$ obey the same equations in the Laplace domain:

$$\hat{E}(s) = -H(s)\,k_{EH}(s) + P(s)\,k_{EP}(s)$$

$$\hat{P}(s) = H(s)\,k_{PH}(s) + R(s)\,k_{PR}(s) + E(s)\,k_{PE}(s)$$

This allowed us to solve for the model parameters analytically in closed form. Further, the cost function of the model for a given dataset at steady state could be easily visualized as subsets of parameters were varied systematically (*Figure 5* and *Figure 8A*; steady state gain was approximated by the 0.5 Hz sinusoidal conditions from Dataset 1, with the gain of efference copy positive feedback fixed at $g = 0.2$). In these plots, for ease of visualization, we display results for the steady state response ($s = 0$), for which the imaginary parts of the filters equal zero. Because the model is linear, all other stimulus conditions can be constructed as linear combinations of the VOR in the dark condition (vestibular stimulus only) and the smooth pursuit condition (visual stimulus only). This gives the following equations for VOR in the dark:

$$E_d(s) = -H_d(s)\,k_{EH}(s) + P_d(s)\,k_{EP}(s)$$

$$P_d(s) = H_d(s)\,k_{PH}(s) + E_d(s)\,k_{PE}(s)$$

and for smooth pursuit:

$$E_p(s) = P_p(s)\,k_{EP}(s)$$

$$P_p(s) = R_p(s)\,k_{PR}(s) + E_p(s)\,k_{PE}(s)$$

where $E_d$ and $P_d$ represent the complex gain and phase of eye velocity and Purkinje cell firing rate, respectively, during VOR in the dark, and $E_p$ and $P_p$ represent the same during smooth pursuit. Solving the model equations above shows that the brainstem parameters are fully constrained, and can be determined directly from the data as follows (equations below hold separately for every value of $s$; for readability, $s$ is not shown for the equations describing the filter parameters below):

$$k_{EH} = \frac{E_p P_d - E_d P_p}{P_p H_d}$$

and

$$k_{EP} = E_p / P_p$$

The parameters describing the inputs to the Purkinje cells ($k_{PH}$, $k_{PE}$, $k_{PR}$) are degenerate, but related to each other by the following equations:

$$k_{PR} = \frac{P_p - k_{PE} E_p}{R_p}$$

$$k_{PH} = \frac{P_d - E_d k_{PE}}{H_d}$$

Thus, given the filter describing one input pathway to Purkinje cells, the other two pathway's filters can be determined. This observation motivated our strategy of fixing the efference copy pathway at different levels of feedback, and observing the requirements this imposes upon the other two pathways.

We give more intuition for this analysis here. As described in the main text, the degenerate direction illustrated in *Figure 5B* indicates that the small Purkinje cell responses observed during the VOR in the dark could reflect either a small vestibular input ($k_{PH}$) alone, or a large vestibular input offset by a large efference copy feedback ($k_{PE}$). However, the full analysis here shows that the degeneracy is actually between all three input pathways to Purkinje cells (vestibular, efference copy, and visual). This degeneracy arises because there are three 'unknowns' (the three input filters to Purkinje cells) but only two independently controllable variables: the vestibular and visual target stimuli. Eye velocity is determined by how these inputs are processed through the circuit; thus, it is not an independently controllable variable. In contrast, there are only two unknown input filters to the brainstem, and thus they are fully constrained by the two independently controllable variables alone.

We also analyzed the model with a Purkinje cell 'stimulation' condition conducted with the head stationary in the dark to mimic experiments in which Purkinje cells were electrically stimulated (*Figure 8A*). This provides one additional constraint, making the model nondegenerate when the stimulation condition is included. For this condition, the Purkinje cell equation was modified to include an additional input $S(s)$ that was only delivered to Purkinje cells:

$$P_s\left(s\right) = E_s\left(s\right) k_{PE}\left(s\right) + S\left(s\right)$$
$$E_s\left(s\right) = P_s\left(s\right) k_{EP}\left(s\right)$$

During this condition, the Purkinje cell response derived from these two equations is:

$$P_s\left(s\right) = S\left(s\right) / \left(1 - k_{EP}\left(s\right) k_{PE}\left(s\right)\right)$$

## Simulation of paradoxical changes

To simulate the 'paradoxical' changes in neural activity during VOR cancellation and smooth pursuit following learning (*Miles and Lisberger, 1981*; *Lisberger, 1994a*; *Figure 7A and E*), we expanded our model to create a population of 20 Purkinje cells whose average activity matched the model fits in the remainder of the paper. For each 'cell', $k_{PH}$ and $k_{PR}$ were scaled by a random factor drawn from a normal distribution $N(1, \sigma)$ with $\sigma = 0.25$ and $0.4$, respectively, for the No Feedback model, and $\sigma = 0.1$ and $20$ for the Strong Feedback model. These random scale factors were normalized to ensure that their average across the population for each model was exactly equal to one, which was necessary to ensure stability in the Strong Feedback model. We then examined the predicted Purkinje cell activity during smooth pursuit and during VOR cancellation before and after simulated learning. The amplitude of the Purkinje cell responses during VOR cancellation (sinusoidal stimulation at 0.5 Hz) was taken as the 'Head Sensitivity' of the cell, mimicking the process used to estimate head sensitivity experimentally (*Lisberger, 1994a*; *Miles and Lisberger, 1981*). Similarly, to simulate the process used to estimate 'Eye Sensitivity' experimentally, the amplitude of the Purkinje cell responses to visual input alone was measured. 'Head Sensitivity' was plotted against 'Eye Sensitivity' to determine whether

each model could replicate the previous finding of apparent changes in Purkinje cell 'Head Sensitivity' (for a given 'Eye Sensitivity') with changes in VOR gain.

## Transient stimulation

To simulate the effect of transient perturbation of Purkinje cells while the head was stationary in the dark (*Lisberger, 1994a*; *Figure 8B and C*), we added an electrical stimulation signal to the model Purkinje cell, and then allowed Purkinje cell activity and eye velocity to evolve according to the model dynamics. *Equation 2* above was modified as follows:

$$\hat{P}\left(t\right) = \left(H * k_{PH}\right)\left(t\right) + \left(E * k_{PE}\right)\left(t\right) + S\left(t\right),$$

where $H(t)=0$ for stimulation in the dark and $S(t)$ is the electrical stimulation input, equal to 1 during the stimulation period of 25 ms in duration and 0 otherwise. For the plots in *Figure 8C*, the model eye velocity for each efference copy feedback strength was scaled to be of equal maximum amplitude.

## Acknowledgements

We thank Jay Bhasin for comments on the manuscript, and Salomon Muller and Emre Aksay for helpful discussions. Funding: This work was supported by the Helen Hay Whitney Foundation Fellowship, National Science Foundation grants DGE-114747 and 0801700, and the Stanford University DARE Doctoral Fellowship Program (HLP), NIH R01 NS104926 (MSG), R01 DC004154 (JLR and MSG), R01 NS072406 (JLR), and a Simons Collaboration on the Global Brain grant (JLR and MSG). In preparing this manuscript, we made a conscious effort to address citation bias. Following the approach outlined in *Dworkin et al., 2020*, we used open source code to assess the gender and racial balance of our citations based on the first names of the first and last authors (*Zhou et al., 2022*). Excluding self-citations to the first and last authors of our current paper, our references contain: 8.82% woman (first author)/woman (last author), 7.13% man/woman, 15.69% woman/man, and 68.36% man/man; and 9.75% author of color /author of color, 17.36% white author/author of color, 24.48% author of color/white author, and 48.41% white author/white author. Due to the historical nature of our inquiry, we also analyzed the gender balance of our citations published after the year 2000, and this subset of references contain: 16.07% woman/woman, 10.71% man/woman, 17.86% woman/man, and 55.36% man/man.

## Additional information

### Competing interests

Jennifer L Raymond: Reviewing editor, eLife. The other authors declare that no competing interests exist.

### Funding

| Funder | Grant reference number | Author |
| --- | --- | --- |
| Simons Foundation | Simons Collaboration on the Global Brain | Mark S Goldman |
| National Institutes of Health | NS072406 | Jennifer L Raymond |
| National Institutes of Health | DC004154 | Mark S Goldman |
| National Institutes of Health | NS104926 | Mark S Goldman |
| National Science Foundation | Graduate Research Fellowship DGE-114747 | Hannah L Payne |
| National Science Foundation | 0801700 | Hannah L Payne |

| Funder | Grant reference number | Author |
|---|---|---|
| Helen Hay Whitney Foundation | | Hannah L Payne |

The funders had no role in study design, data collection and interpretation, or the decision to submit the work for publication.

## Author contributions

Hannah L Payne, Conceptualization, Software, Formal analysis, Investigation, Visualization, Methodology, Writing – original draft, Writing – review and editing; Jennifer L Raymond, Conceptualization, Data curation, Supervision, Funding acquisition, Investigation, Methodology, Writing – original draft, Writing – review and editing; Mark S Goldman, Conceptualization, Supervision, Funding acquisition, Investigation, Methodology, Writing – original draft, Writing – review and editing, Formal analysis

## Author ORCIDs

Hannah L Payne ⓘ http://orcid.org/0000-0003-4625-5706
Jennifer L Raymond ⓘ http://orcid.org/0000-0002-8145-747X
Mark S Goldman ⓘ https://orcid.org/0000-0002-8257-2314

## Decision letter and Author response

Decision letter https://doi.org/10.7554/eLife.84770.sa1
Author response https://doi.org/10.7554/eLife.84770.sa2

# Additional files

## Supplementary files

• MDAR checklist

## Data availability

Data and code are available at: https://github.com/goldman-lab/VOR-model-Payne (copy archived at *Payne, 2024*). Data is also available at: https://doi.org/10.5061/dryad.rr4xgxdg6.

The following dataset was generated:

| Author(s) | Year | Dataset title | Dataset URL | Database and Identifier |
|---|---|---|---|---|
| Payne H, Goldman M, Raymond J | 2024 | Data for: Interactions between circuit architecture and plasticity in a closed-loop cerebellar system | https://doi.org/10.5061/dryad.rr4xgxdg6 | Dryad Digital Repository, 10.5061/dryad.rr4xgxdg6 |

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
