## [Editor Report]

Payne et al. present a novel model that predicts sites and directions of plasticity within the vestibular cerebellum to explain the basis for learned adjustments to reflexive eye movements in monkeys. The model is convincingly constrained by prior biological observations and makes an important prediction about the level of feedback available to the cerebellar cortex and how this level determines the plasticity required to explain post-learning changes in activity. Overall, a number of exciting and testable experiments will likely be motivated by this study.

---

## [Decision Letter]

**Decision letter after peer review:**

Thank you for submitting your article "Interactions between circuit architecture and plasticity in a closed-loop system" for consideration by *eLife*. Your article has been reviewed by 3 peer reviewers, and the evaluation has been overseen by a Reviewing Editor and Michael Frank as the Senior Editor. The reviewers have opted to remain anonymous. We apologize for the extraordinary delay in completing the review process.

Essential revisions:

1) Many of the concerns from all 3 reviewers center around the strength of evidence for weak feedback, which is a fundamental conclusion of the paper. Please pay particular attention to the questions and comments below relating to this point.

2) The reviewers have suggested a number of ways that the paper's claims should be placed more appropriately within the existing literature.

3) Questions were raised about how the model performance might change after learning, which should be addressed.

4) Some clarification of 'feedback' as carried by mossy/ parallel fiber pathways vs. climbing fiber pathways will be important for a general audience.

*Reviewer #1 (Recommendations for the authors):*

1. The paper seems to be written as a refutation of Miles and Lisberger (1981), but what is really striking here are actually the similarities with that paper. The main proposal by Miles and Lisberger was that plasticity in the brainstem (under control of Purkinje cell input from the flocculus) was primarily responsible for VOR learning. In the current manuscript, learning is always associated with exactly the kind of brainstem plasticity proposed by Miles and Lisberger (shown in Figure 6D and stated explicitly in lines 349-351, "the correct learned increases in Purkinje cell activity during VOR cancellation depended on the existence of plasticity in the direct pathway through the brainstem"). In that sense, a fundamental conclusion of both papers is that brainstem plasticity is key for VOR adaptation. I am concerned that as written, the paper may give the erroneous impression that it is overturning the Miles and Lisberger model, when really, it seems to be solidly consistent with it. The more generalizable conclusion here would seem to be that feedback gain can lead to counterintuitive effects on the relationship between plasticity, neural activity, and behavior – which of course, is exactly the point of both Miles and Lisberger and the Lisberger 1994 papers, even if the current manuscript extends this idea and tests some aspects of it more explicitly.

If I'm not mistaken, it was only later that the Lisberger models (1992 and 1994) incorporated an additional site of plasticity in the cerebellar cortex, which is where the question of the directionality of potentiation/ depression of vestibular inputs to Purkinje cells that the current manuscript focuses on became an issue. Clarification of this point would be important, to make it more clear what aspects of the previous models are being explored/ refuted here. Again, in my view, the data are generally strikingly consistent with those earlier models and in fact the results of the Lisberger 1994 model are reproduced several times here. (As a side note, there are 4 Lisberger 1994 J Neurophys papers but only one is cited in the reference list – please check whether some of the references are meant to be to the other papers, for example when cited as Lisberger et al. rather than single author.)

2. Figure 3 nicely shows the model fits to data for Purkinje cell and eye velocity traces across a range of tracking conditions. However, I had two issues with this.

a. Why are the fits to data shown only before learning? Fits to the data after learning would seem to be just as crucial.

b. As I understand it, a key argument provided by the current model is that the strength of efference copy feedback can't be inferred from neural recordings and behavior alone, when visual inputs (retinal and target) to Purkinje cells are considered in addition to vestibular and eye velocity inputs. The inclusion of the retinal slip, and especially, the target prediction signal, are the fundamental difference from the Lisberger 1994 model. The effects of these differences on behavioral and neural responses to pursuit stimuli, which are currently limited to sinusoidal tracking, should be more carefully explored, both before and after learning. In particular, eye acceleration responses to step-ramp pursuit stimuli are crucial. For instance, it would seem to me that the explanation provided by the authors, that the visual feedback loop "works harder" after learning (Line 324) might bring with it some changes in step-ramp pursuit after learning. This should be tested in the model.

3. The main evidence for low levels of feedback is presented in Figure 8, but the models are plotted separately from the experimental data, and there is no quantitative comparison, making it hard to judge/compare the goodness of fits. In particular it appears that the model is qualitatively consistent with the experimental data within a range of feedback strengths, including some that are higher (up to at least 0.6, and possibly up to 0.8, which is not shown) than the switch from net depression to net potentiation shown in Figure 6C (net change).

4. There is other evidence besides the VOR work cited here that argues for the existence of substantial efference copy inputs to the cerebellum on the mossy fiber pathway, for example Giovannucci et al., Nature Neuroscience 2017 for eyeblink conditioning. This should be addressed.

5. The Lisberger 1994 papers focus extensively on altered dynamics of VOR responses following learning in low and high gain conditions. From Figure 6. Supp 1 it appears that those are not explored/ reproduced here. Is that correct? If so it should be mentioned as a potential caveat.

*Reviewer #2 (Recommendations for the authors):*

The model performs best when feedback to the cerebellar cortex is weak. This conclusion is supported by the evidence presented in Figure 8 which was obtained in animals that hadn't undergone VOR learning. Is it possible that feedback strength changes during learning? If so, how would this effect model performance?

Is weak feedback to the cerebellar cortex a universal principle of the cerebellar function? Are there learning regimes – either in the oculomotor system or not- where potentiation occurs because feedback is strong? Said another way, is the strength of feedback a determinate whether PC firing is increased or decreased as a result of leaning-induced plasticity. Without a wider scope to the context of this finding, the authors seem to be studying a somewhat esoteric problem.

The results are aimed at explaining the results of Miles and Lisberger (1981) regarding potentiation vs depression at vestibular inputs onto PCs. However, more recently, and perhaps more controversially, the De Zeeuw group has proposed that LTP, not LTD, at parallel fiber to PC synapses explains gain-increase learning because floccular PCs are "upward-bound". Does the model similarly rule out these findings of the De Zeeuw group as well?

*Reviewer #3 (Recommendations for the authors):*

1. (Expanded from weakness 1) It would be important to make the distinction between feedback for dynamics (as in the current model) and feedback for learning (as would be mediated by climbing fibers) clear and discuss what it means for the conclusions drawn in this study. Indeed, a role of feedback in learning is already implicitly assumed by the authors when jointly fitting the model before and after learning. I think it would help the reader to make this clear from the outset -- maybe adding distinct but related schematics for the two feedback conditions would help. Indeed the general conclusion that feedback (in general) is weak may only apply to the first view (i.e. dynamics). This distinction has also been recently studied in the context of cerebrum-cerebellar models (feedback for learning: Boven et al. Nature Comms 2023 and for dynamics: Pemberton et al. bioRxiv 2022), would be useful to discuss how your conclusions contrast with these studies.

2. (Expanded from weakness 2) There are some potential limitations of the conclusions drawn due to the model inference methods used. The methods used (fmincon) can easily get stuck in local minima and more importantly they do not provide an overview of the likelihood of parameters given the data. A few studies have now shown that it is important to apply more powerful inference techniques both to infer plasticity (Costa et al. Frontiers 2013, Bykowska et al. Frontiers 2019) and neural dynamics (Gonçalves et al. *eLife* 2020). As highlighted by Costa et al. Frontiers 2013 using more standard fitting methods can easily lead to misleading "best parameters". Maybe this is not an issue with the present model, but it would be important to discuss why and ideally show that irrespective of the initial conditions of the model parameters (i.e. before fitting) the fitting method always produces very similar best-fit parameters.

3. There is a lack of clarity on how to best interpret inhibitory/excitatory pathways. For example, should we interpret the efference copy feedback as inhibiting or exciting PC activity? How would this pathway be implemented in the brain? The pathway is only briefly described in section 2.1 amongst the other filters; more emphasis should be placed on it to help guide the reader. You could also mention k_{pe} at the start of section 2.2 to make it clear what it is.

4. The closed-form analysis in section 8.4 is useful but super intuitive to me. Why is there a degeneracy in how the Purkinje cell inputs are shaped and not in the brainstem inputs? Would it be possible to provide a more accessible explanation in the text? For example, I find the right plot in figure 6D surprising.

5. It is unclear to me *why* feedback might be important for the brain from a functional point of view (i.e. what are the advantages of no feedback vs strong feedback)? I think the paper would benefit from more discussion on this point, even if only from a more speculative angle.

---

## [Author Response]

Essential revisions:1) Many of the concerns from all 3 reviewers center around the strength of evidence for weak feedback, which is a fundamental conclusion of the paper. Please pay particular attention to the questions and comments below relating to this point.

We have significantly revised the text to clarify the evidence for weak feedback. We now emphasize that the strongest evidence for weak efference copy feedback comes from comparing the model results to previous experimental evidence for error-driven plasticity in the cerebellar cortex (e.g. LTD of parallel fiber inputs to Purkinje cells that are coactive with climbing fibers). This work was motivated by the puzzle that previous experimental evidence for such plasticity was incompatible with a leading circuit model of VOR learning, the Miles-Lisberger model. Our results show that only models with weak or no efference copy feedback predicted synaptic changes compatible with error-driven plasticity in the cerebellar cortex. Additionally, we show that these models explain all other available data, including paradoxical changes in neural activity during closed-loop behavioral paradigms. The perturbation analysis in Figure 8 is an orthogonal test of circuit feedback strength that complements this conclusion.

2) The reviewers have suggested a number of ways that the paper's claims should be placed more appropriately within the existing literature.

We have edited the text to better situate the results in the existing literature. We now discuss how a similar strategy could be applied to other systems with feedback, both within and outside of the cerebellum. The exact implications of feedback for inferring plasticity from neural activity recordings will depend on the specifics of each circuit. In fact, even within the VOR circuit, we discuss how the distribution of plasticity between the cerebellar cortex and in the brainstem could vary over the time course of learning as memories are consolidated.

Connecting to other cerebellar-dependent learning paradigms, we note that there has been a seeming disconnect between the plasticity mechanisms suggested for VOR learning and other learning paradigms such as eye-blink conditioning. We now clarify how our model reconciles these different paradigms, suggesting that similar cerebellar plasticity mechanisms may be operating across different tasks and cerebellar subregions. We also situate the result in the context of other work that supports specific plasticity mechanisms other than parallel fiber-Purkinje cell LTD, such as LTP of parallel fibers carrying oppositely directed vestibular input, and we discuss the functional equivalence of these mechanistic variations.

We also now clarify, and include additional, references to key experimental studies by Lisberger (1994), and to other approaches for analyzing “sloppy” model parameters.

3) Questions were raised about how the model performance might change after learning, which should be addressed.

We have clarified and expanded the discussion of several points related to how model performance might change after learning. First, we clarify that the dataset fit after learning includes a wide range of behavioral data, and a more restricted set of neural data due to limited data availability. Overall, a key strength of this study is that it pulls together data from many studies in order to capture circuit dynamics both before and after learning. Second, we discuss the implications of plasticity in the efference copy feedback pathway, which was held fixed in the model. Third, we clarify how visual pathways can “work harder” (i.e. increase their activity) during VOR cancellation after learning, without undergoing plasticity.

4) Some clarification of 'feedback' as carried by mossy/ parallel fiber pathways vs. climbing fiber pathways will be important for a general audience.

We now explicitly state in the text that we have focused on studying feedback that affects circuit dynamics, rather than feedback for learning. We clarify that we do not explicitly model how climbing fiber feedback instructs plasticity, and fully agree that this form of feedback is essential for learning.

Reviewer #1 (Recommendations for the authors):1. The paper seems to be written as a refutation of Miles and Lisberger (1981), but what is really striking here are actually the similarities with that paper. The main proposal by Miles and Lisberger was that plasticity in the brainstem (under control of Purkinje cell input from the flocculus) was primarily responsible for VOR learning. In the current manuscript, learning is always associated with exactly the kind of brainstem plasticity proposed by Miles and Lisberger (shown in Figure 6D and stated explicitly in lines 349-351, "the correct learned increases in Purkinje cell activity during VOR cancellation depended on the existence of plasticity in the direct pathway through the brainstem"). In that sense, a fundamental conclusion of both papers is that brainstem plasticity is key for VOR adaptation. I am concerned that as written, the paper may give the erroneous impression that it is overturning the Miles and Lisberger model, when really, it seems to be solidly consistent with it. The more generalizable conclusion here would seem to be that feedback gain can lead to counterintuitive effects on the relationship between plasticity, neural activity, and behavior – which of course, is exactly the point of both Miles and Lisberger and the Lisberger 1994 papers, even if the current manuscript extends this idea and tests some aspects of it more explicitly.If I'm not mistaken, it was only later that the Lisberger models (1992 and 1994) incorporated an additional site of plasticity in the cerebellar cortex, which is where the question of the directionality of potentiation/ depression of vestibular inputs to Purkinje cells that the current manuscript focuses on became an issue. Clarification of this point would be important, to make it more clear what aspects of the previous models are being explored/ refuted here. Again, in my view, the data are generally strikingly consistent with those earlier models and in fact the results of the Lisberger 1994 model are reproduced several times here. (As a side note, there are 4 Lisberger 1994 J Neurophys papers but only one is cited in the reference list – please check whether some of the references are meant to be to the other papers, for example when cited as Lisberger et al. rather than single author.)

We thank the reviewer for their comments and agree that our conclusions do not refute the Miles and Lisberger hypothesis in its entirety. Indeed, we find that a site of plasticity in the brainstem is critical to explain the “paradoxical” increase in Purkinje cell activity during VOR cancellation after learning. Thus our results corroborate the existence of brainstem plasticity in the Miles and Lisberger hypothesis, although in our model this site was required regardless of the strength of efference copy feedback.

However, while we found that brainstem plasticity was necessary to explain Purkinje cell activity during VOR cancellation, we disagree that brainstem plasticity is key for VOR adaptation in general. In particular, we predict that at different stages of systems consolidation, different distributions of plasticity between the cerebellar and brainstem sites would be expected—including potentially no plasticity in the brainstem early in learning. We have added a note pointing the reader to the relevant section of the Discussion (“Implications for systems consolidation of cerebellar learning”) on line 388.

We also feel that the site of plasticity in the cerebellar cortex is critical to the debate. In contrast to the Miles-Lisberger hypothesis (see clarification below), we find that the observed increase in Purkinje cell activity during VOR cancellation is fully compatible with net depression of vestibular inputs to the cerebellar cortex. We thus frame this work as a reconciliation of the Lisberger model, which was motivated by extensive neural and behavioral data, with elements of the earlier Marr-Albus-Ito model and—most importantly—with evidence that plasticity in the cerebellar cortex is guided by error signals (i.e. retinal slip). The Lisberger model, and our implementation of it (the Strong Feedback model) is incompatible with this error-driven plasticity mechanism, which is our main evidence for a model with weaker positive feedback. Additionally, even though we reproduce the results of Lisberger (1994) in several figures, we ultimately find that this model is additionally incompatible with the response to transient perturbation, discussed further in Point 3 below. We agree that the similarities and differences with the Miles and Lisberger model should be more explicit in the manuscript, and have now incorporated this in Section 3.0 lines 440-457*.*

Regarding the question of whether Miles and Lisberger (1981) incorporated a site of plasticity in the cerebellar cortex: they conclude that the strength of vestibular input to Purkinje cells increases with learning, opposite to the decrease predicted by the Ito hypothesis. They remain agnostic as to whether the increase in this vestibular input is due to plasticity at a single site, presynaptic to both the flocculus and the brainstem (site C in their Figure 3), or at two separate sites, one in the brainstem and one in the flocculus (sites A and D in their Figure 3), but both possibilities result in a net increase in the strength of vestibular input to the cerebellar cortex. This interpretation is supported by their theoretical argument: “…any changes in the "gain" of the brainstem vestibular signal should be matched by corresponding changes in the "gain" of the flocculus vestibular signal” (Miles and Lisberger 1981), and their interpretation of experimental results using the VOR cancellation paradigm: “Single unit recordings in the primate flocculus have revealed that the average strength of the head velocity (vestibular) signals carried by the P-cells varies with the gain of the VOR” (Miles and Lisberger 1981).

To clarify which of the four Lisberger 1994 papers are cited, we originally cited two relevant Lisberger 1994 papers: (1) Lisberger, Pavelko, Bronte-Stewart, and Stone 1994, “Neural basis for motor learning in the vestibuloocular reflex of primates. II. Changes in the responses of horizontal gaze velocity Purkinje cells in the cerebellar flocculus and ventral paraflocculus” and (2) Lisberger 1994, “Neural basis for motor learning in the vestibuloocular reflex of primates. III. Computational and behavioral analysis of the sites of learning”. Note that (2) appears several items before (1) in the manuscript reference list, which is sorted according to APA style such that among references with the same first author, those with a single author are listed before those with multiple authors regardless of year. For completeness, we have now additionally included a reference to Lisberger, Pavelko, Broussard 1994, “Neural basis for motor learning in the vestibuloocular reflex of primates. I. Changes in the responses of brain stem neurons.”

2. Figure 3 nicely shows the model fits to data for Purkinje cell and eye velocity traces across a range of tracking conditions. However, I had two issues with this.a. Why are the fits to data shown only before learning? Fits to the data after learning would seem to be just as crucial.

Thank you for the question about why the model fits before learning shown in Figure 3 are not replicated after learning. Figure 3 includes both neural and behavioral data before learning. Unfortunately, neural data for the full range of stimulus conditions shown before learning do not exist after learning, to our knowledge. Therefore, the neural data used to constrain the model after learning were restricted to the steady state and transient changes in Purkinje cell modulation during steps (Lisberger, Pavelko, Bronte-Stewart, and Stone 1994) and the amplitude during low-frequency sines waves (Watanabe 1985) in vestibular input alone (see Section 8.1 Materials and methods, Data set). The model Purkinje cell response during the VOR after learning is shown in Figure 6—figure supplement 4A,D. Since the models merely had to replicate a scalar change in Purkinje cell activity, which they each did perfectly, we did not feel that this result merited inclusion in a main figure.

However, we do show the model fits to *behavioral data* after learning for the wide range of VOR frequencies (0.5 to 50 Hz) tested by Lisberger and Ramachandran 2005 in Figure 6A,B. Since our model is linear, this wide frequency range is sufficient to construct the predicted response to other arbitrary stimuli, such as the steps in head velocity input shown in Figure 6—figure supplement 1.

We have modified the titles in Figure 6A,B to emphasize that we are comparing the dynamics of VOR behavior after learning between models and data. Additionally, we modified Section 2.3 to clarify the limitation of the data: “Each model was fit to learned changes in behavior during the VOR in the dark across a broad range of stimulus frequencies from 0.5 Hz to 50 Hz (data from Ramachandran and Lisberger, 2005; Figure 6A) as well as to learned changes in Purkinje cell activity during the VOR at low frequencies (data from Lisberger et al., 1994; Watanabe, 1985). We note that we were limited to analyzing only low frequency neural data, since neural data for the full range of stimulus conditions analyzed before learning does not exist for after learning.”

b. As I understand it, a key argument provided by the current model is that the strength of efference copy feedback can't be inferred from neural recordings and behavior alone, when visual inputs (retinal and target) to Purkinje cells are considered in addition to vestibular and eye velocity inputs. The inclusion of the retinal slip, and especially, the target prediction signal, are the fundamental difference from the Lisberger 1994 model. The effects of these differences on behavioral and neural responses to pursuit stimuli, which are currently limited to sinusoidal tracking, should be more carefully explored, both before and after learning. In particular, eye acceleration responses to step-ramp pursuit stimuli are crucial. For instance, it would seem to me that the explanation provided by the authors, that the visual feedback loop "works harder" after learning (Line 324) might bring with it some changes in step-ramp pursuit after learning. This should be tested in the model.

A key difference from the Miles-Lisberger model is that we include a retinal slip pathway that is flexibly fit to data. The Lisberger 1994 model did include a retinal slip pathway, but this pathway was assumed to represent the command for eye acceleration during pursuit and was not fit in the full model. We demonstrate that when the retinal slip pathway is fit in the context of the closed-loop circuit model, it changes from acceleration-like to velocity-like as the strength of efference copy feedback decreases. This allowed models with weak or no efference copy feedback to explain all available data.

As the reviewer points out, an additional difference is our inclusion of a visual target prediction signal. This signal was required (1) to obtain the best quantitative fits to the data before learning, (2) to account for the observation that smooth pursuit eye movements continue even if an object disappears behind an occluder, and (3) to account for the observation that the gain of eye movements during low-frequency visual-vestibular stimuli (e.g. VOR cancellation) after learning is unchanged (Guo and Raymond 2010). However, note that without this visual prediction signal, the main results -- the sites and signs of plasticity in each model, and the paradoxical increases in

Purkinje cell firing during VOR cancellation -- were unchanged (see Figure 7—figure supplement 1). Thus, the target prediction pathway was necessary to explain all the experimental details, but the retinal slip pathway alone was sufficient to understand how plasticity and internal feedback interact in the circuit.

We also have clarified our confusing wording about the visual pathways “working harder”, by which we mean that there is more input to Purkinje cells from this pathway after learning. Given that these visual pathways “work harder” to cancel the VOR after learning, the reviewer raises an interesting question about how learning affects smooth pursuit. Lisberger (1994) tested the effect of VOR learning on the initiation of pursuit, and concluded that there is “little to no consistent effect on pursuit eye movements” after VOR learning. Accordingly, we did not include plasticity in either visual pathway in our model, and thus there is no change in smooth pursuit performance after learning.

How do the visual pathways “work harder” if they do not undergo plasticity? There are two ways in which the visual pathways can work harder to cancel the VOR, depending on whether or not the visual target prediction pathway is included:

If only retinal slip is included, then the performance of VOR cancellation will worsen after learning, since it is harder to reduce/cancel the gain of the VOR after having just learned to increase it. This worsened performance leads to increased retinal slip, thus increasing the activity of the visual pathway, which increases Purkinje cell firing during ipsiversive head turns, counteracting the reduction in firing due to LTD of vestibular inputs. In effect, the retinal slip works to partially reduce visual error through negative feedback via Purkinje cells, which produces the observed “paradoxical” increase in Purkinje cell firing.If a visual prediction pathway is included, this pathway adjusts its contribution to the eye movement command “on-the-fly” in order to increase Purkinje cell firing and achieve correct cancellation. We assume that this adjustment can occur through cortical mechanisms that do not require plasticity.

In either case, even though the visual pathways increase their activity during VOR cancellation after learning, they do not undergo plasticity in our model, thus there is no change in smooth pursuit performance, whether to sinusoidal or step-ramp stimuli, after learning. Note that we discuss the possibility of plasticity of the visual pathways in the Discussion. Clarification of the fact that there is no plasticity in the visual pathways and this more extensive discussion has now been incorporated into the text (Section 2.4, lines 362-375).

3. The main evidence for low levels of feedback is presented in Figure 8, but the models are plotted separately from the experimental data, and there is no quantitative comparison, making it hard to judge/compare the goodness of fits. In particular it appears that the model is qualitatively consistent with the experimental data within a range of feedback strengths, including some that are higher (up to at least 0.6, and possibly up to 0.8, which is not shown) than the switch from net depression to net potentiation shown in Figure 6C (net change).

The reviewer is correct that the model responses to transient stimulation are not quantitatively compared to the data in Figure 8. This was done intentionally: first, since the time course of data from Lisberger 1994 was only available for a single monkey, and is from a different study than those used to fit our model, we did not want to overstep the quantitative conclusions that can be drawn. Second, we note that the relationship between feedback strength and the time constant (“*τ*”) of eye velocity decay following perturbation is nonlinear. The relationship is given by *τ* ∝ 1/(1-feedback), which we now show in an additional panel (Figure 8D). Thus, large changes in time constant only appear for feedback strengths approaching unity, and the minor changes in *τ* for weaker feedback strengths make it impossible to quantitatively disambiguate between weak feedback models. However, the model of Miles and Lisberger 1981 requires a net positive feedback strength of 1, which is inconsistent with the simulations in Figure 8. We have also updated Figure 8C to now show the model prediction for feedback = 0.8, which also appears inconsistent. This being said, overall, we agree with the reviewer that Figure 8 is consistent with a range of feedback strengths up to approximately 0.6. We indicate the uncertainty around this measurement as follows: “Although the precise strength of the efference copy feedback is difficult to ascertain because the time constant of decay of responses changes only gradually until the gain of the feedback loop approaches one (Figure 8D), this suggests that feedback in this circuit is relatively weak (Figure 8E; see Discussion).”

Regardless of the exact match between data and model in Figure 8, however, we emphasize that the primary evidence for weak feedback strength comes from comparing our results with other studies suggesting net depression at this site during learned increases in the VOR. Our results show a strict relationship between feedback strength and the direction of plasticity at the vestibular inputs to the cerebellar cortex (kPH) (Figure 6C), and reconcile net depression at this site with the surprising increases in Purkinje cell response observed during VOR cancellation after VOR-increase learning (Figure 7). We have updated the text to indicate that the perturbation analysis complements the main evidence for relatively weak feedback, namely that only models with weak feedback were compatible with error-driven plasticity in the cerebellar cortex (Section 2.5)

4. There is other evidence besides the VOR work cited here that argues for the existence of substantial efference copy inputs to the cerebellum on the mossy fiber pathway, for example Giovannucci et al., Nature Neuroscience 2017 for eyeblink conditioning. This should be addressed.

We agree that there is substantial evidence for efference copy inputs to the cerebellum, and thank the reviewer for bringing our attention to Giovannucci et al. 2017. This study finds that granule cell activity matches the temporal dynamics of learned eyeblink responses, consistent with an efference copy of the learned motor response being present in granule cells. However, there is a critical distinction between an efference copy input, and efference copy input configured specifically as a positive feedback loop through the cerebellar cortex. An efference copy pathway may be present (indeed, there is substantial anatomical evidence for regions that both affect eye movements and send input to the cerebellar flocculus), but that does not mean that this input must be configured as a positive feedback loop where eye movement commands driven by Purkinje cells feed back to affect the same Purkinje cells, as proposed in the Miles-Lisberger hypothesis. Additionally, the functional strength of feedback is impossible to determine from anatomy alone. We illustrate this in Figure 8F, but have now also expanded and emphasized this point in the text and included the reference to Giovannucci et al. 2007 (Discussion, Section 3.1, lines 480-490).

5. The Lisberger 1994 papers focus extensively on altered dynamics of VOR responses following learning in low and high gain conditions. From Figure 6. Supp 1 it appears that those are not explored/ reproduced here. Is that correct? If so it should be mentioned as a potential caveat.

Our model reproduced the altered dynamics of VOR behavioral responses following learning, as quantified by the gain and phase of the VOR before and after learning across a wide range of frequencies (Figure 6A,B). Since our model is linear, this wide frequency range is sufficient to construct the predicted response to other arbitrary stimuli including steps (Figure 6—figure supplement 1), which do indeed show changes in steady state and transient components in the same direction as measured by Lisberger and Pavelko (1986). This transient component was not quantitatively compared since it depends on the exact dynamics of the imperfect step stimulus and on the method of quantification (max taken for each cell individually v. for the population). We have now updated the labels and the color scheme in Figure 6A,B to clarify that these panels show that the model reproduces experimentally observed changes in VOR dynamics after learning, and have clarified the relationship to the main figure and previous work in the legend of Figure 6—figure supplement 1.

Regarding changes in neural dynamics, we also considered the change in the transient Purkinje cell response to steps in head velocity input after VOR learning reported in Lisberger, Pavelko, Bronte-Stewart and Stone 1994. In that study, the transient response was measured as “the peak firing rate in the first 50 ms after the onset of head motion”. This is described in Materials and methods, Section 8.1: “Both the steady-state changes (averaged 100-200 ms after step onset) and transient changes (peak firing rate measured 0-50 ms after step onset) after learning reported by Lisberger et al. were included in the cost function.” The term capturing this transient response was included in the cost function C _PostLearn_ in Section 8.2 as (p^transpost−ptranspost)2 , which “encourages the transient Purkinje cell responses to steps (P_trans^post) to mimic the transient responses after learning reported in Dataset 3 (Lisberger, Pavelko, Bronte-Stewart, et al., 1994).” For the same reasons given above for behavioral transients, this was only a weak constraint and thus we do not quantitatively compare the transient amplitudes of the Purkinje cell step response.

Reviewer #2 (Recommendations for the authors):The model performs best when feedback to the cerebellar cortex is weak. This conclusion is supported by the evidence presented in Figure 8 which was obtained in animals that hadn't undergone VOR learning. Is it possible that feedback strength changes during learning? If so, how would this effect model performance?

The reviewer brings up the interesting possibility that the strength of efference copy feedback may itself change with learning. We did not include plasticity at this synapse in the model because it was not necessary to explain the available data, but it is likely that plasticity does in fact occur. We briefly mentioned this possibility in the Discussion, but have now expanded that section (3.4.1, “Model Assumptions: Pathways Undergoing Plasticity”) to clarify the qualitative predictions the model would make if a plasticity rule compatible with the observed plasticity at k _PH_ (i.e., correlated activity of climbing fibers and parallel fibers induces synaptic depression of those parallel fibers) was applied to the efference copy feedback pathway.

The relevant paragraph is copied here: “We first consider the correlational plasticity rule applied to the efference copy pathway. This pathway (kPE) is driven by ipsiversive eye velocity, whereas climbing fiber activity in the flocculus is driven by contraversive retinal slip and suppressed by ipsiversive slip (Raymond and Lisberger, 1998). During learning to increase the VOR, ipsiversive eye movements are accompanied by ipsiversive retinal slip. Hence, activation of parallel fibers carrying efference copy signals should be accompanied by little or no climbing fiber activity, which in vitro has been shown to induce potentiation of the parallel fibers inputs to Purkinje cells (Crepel and Jaillard, 1991; Suvrathan et al., 2016). Thus, we predict net potentiation of kPE for learned increases in the VOR. Applying similar logic to learned decreases in the VOR predicts net depression of kPE. However, since the amplitude of eye movements is small during training to decrease the VOR, the plastic changes may be smaller in this case. In both cases, the changes in kPE are in the correct direction to support learning of amplified or attenuated eye movements, respectively. Changes in kPE would also lead to alterations of VOR dynamics, as well as amplitude, due to changing the strength of the efference copy feedback loop (Lisberger and Sejnowski, 1992).”

Is weak feedback to the cerebellar cortex a universal principle of the cerebellar function? Are there learning regimes – either in the oculomotor system or not- where potentiation occurs because feedback is strong? Said another way, is the strength of feedback a determinate whether PC firing is increased or decreased as a result of leaning-induced plasticity. Without a wider scope to the context of this finding, the authors seem to be studying a somewhat esoteric problem.

The reviewer is raising the important question of whether our results generalize across the cerebellum. We note first that we are studying the cerebellum to illustrate a core problem in modeling systems throughout the brain: how to disambiguate plasticity in the face of ubiquitous feedback loops, both within the brain and between the brain and the environment. Within the cerebellum, we illustrate a general approach to determine how plasticity is distributed between the cerebellar cortex and the brainstem or deep cerebellar nuclei in the face of potential feedback loops.

The question about how the results generalize across the cerebellum can be split into two parts: (1) Does feedback strength always determine the direction of plasticity in the cerebellar cortex, such that stronger feedback causes a switch from depression to potentiation? (2) If so, are there instances where feedback is strong and sensory inputs therefore potentiate?

Regarding the first question: while there is an anatomical basis for feedback in other cerebellar circuits, such as for eye blink conditioning, the specific effect of feedback will depend on the details of each circuit. The broader implications of this work are not the exact numerical details of the model fits, but the strategy for fitting a closed-loop model to data, in order to analyze the interactions between model architecture and plasticity. The extensive data available from the VOR system allowed us to identify the regime of possible feedback strengths that are compatible with parallel fiber-Purkinje cell LTD. The same strategy could be applied to other cerebellar learning paradigms in order to determine the quantitative relationship between circuit feedback and plasticity of specific input pathways. We have added a short paragraph to the Discussion to emphasize this point (last paragraph of Section 3.1).

Within the oculomotor system, we can speculate about the second question raised above: are there instances where feedback is strong and vestibular sensory inputs potentiate? Our model represents a single time point during learning, but other studies indicate that memory transfers from the cerebellar cortex to the brainstem over time. Taken to the extreme, for the cerebellar cortex to return exactly to its original, pre-learning output, if there is some efference copy input, we actually conclude that some net potentiation of the vestibular inputs to Purkinje cells would actually be required. This is now more fully addressed in the Discussion (Section 3.2, Implications for systems consolidation of cerebellar learning), which concludes: “This provides the possibility that the net sign of plasticity in vestibular inputs to Purkinje cells could be depression at early stages of learning and potentiation after complete consolidation of learning.” More generally, there may be learning regimes, either within the VOR system or elsewhere, where there is sufficiently strong feedback of large changes elsewhere in the circuit to produce potentiation of sensory inputs that would otherwise be expected to depress.

The results are aimed at explaining the results of Miles and Lisberger (1981) regarding potentiation vs depression at vestibular inputs onto PCs. However, more recently, and perhaps more controversially, the De Zeeuw group has proposed that LTP, not LTD, at parallel fiber to PC synapses explains gain-increase learning because floccular PCs are "upward-bound". Does the model similarly rule out these findings of the De Zeeuw group as well?

The reviewer is referring to a proposal by the de Zeeuw group in which adaptive changes in Purkinje cell activity during VOR learning are driven by LTP of parallel fiber inputs that are active during contraversive head motion, rather than LTD of ipsiversive inputs as originally proposed by Ito and Kano (de Zeeuw 2021; Schonewille et al. 2011). This is motivated in the context of cerebellar “microzones” with either lower or higher spontaneous firing rates, which are proposed to achieve learning primarily through LTP-driven increases or LTD-driven decreases in firing rate, respectively.

Our findings do not fundamentally contradict the de Zeeuw proposal. Our model is agnostic to the biological mechanism by which the vestibular inputs to Purkinje cells undergo net “depression”, defined in our paper as a net reduction in the postsynaptic response to ipsiversive vestibular input. Specifically, instead of requiring LTD of the parallel fiber-Purkinje cell synapse during ipsiversive head turns, such depression could require LTP of the parallel fiber-molecular layer interneuron synapse, or LTP of the interneuron-Purkinje cell synapse.

Additionally, for the eye movement response, the net impact of LTD of ipsiversive vestibular inputs should be identical to the impact of LTP of contraversive vestibular inputs. There is one cerebellar flocculus in each hemisphere, with each receiving the appropriate copy of vestibular input, and each affecting eye movements. Motor output depends on the difference in output between the two hemispheres, relative to their baselines. To give a concrete example: a decrease in activity in the right hemisphere during rightward head turns (LTD of ipsiversive inputs) would cause the eyes to move faster to the left, identical to an increase in activity in the left hemisphere during the same rightward head turns (LTP of contraversive inputs).

This question has now been addressed in the text in a new section of the Discussion, Section 3.4.2, “Implementation of error-driven plasticity in cerebellar cortex”.

Reviewer #3 (Recommendations for the authors):1. It would be important to make the distinction between feedback for dynamics (as in the current model) and feedback for learning (as would be mediated by climbing fibers) clear and discuss what it means for the conclusions drawn in this study. Indeed, a role of feedback in learning is already implicitly assumed by the authors when jointly fitting the model before and after learning. I think it would help the reader to make this clear from the outset -- maybe adding distinct but related schematics for the two feedback conditions would help. Indeed the general conclusion that feedback (in general) is weak may only apply to the first view (i.e. dynamics). This distinction has also been recently studied in the context of cerebrum-cerebellar models (feedback for learning: Boven et al. Nature Comms 2023 and for dynamics: Pemberton et al. bioRxiv 2022), would be useful to discuss how your conclusions contrast with these studies.

We fully agree with the reviewer that our conclusions do not preclude an important role for many other types of feedback, including as an instructive signal for learning. Instead of explicitly considering feedback for learning in our model, we consider static snapshots before and after learning to infer plasticity, while remaining agnostic to the neural algorithm used to achieve such plasticity. A widely held hypothesis is that motor error signals carried by climbing fibers instruct LTD at co-active parallel fiber inputs to Purkinje cells; this is indeed a form of feedback, operating on a slower timescale than “feedback for dynamics.” This “feedback for learning” is not modeled here but is fully consistent with our results, as discussed in a new paragraph of our Discussion (end of Section 3.4.1 “Pathways undergoing plasticity”).

Note that even within the category of “feedback for dynamics”, our specific conclusion that strong feedback is unnecessary only applies to the efference copy feedback loop during VOR learning. In fact, one of our conclusions is that it is critical to consider external feedback of visual motion from the environment in order to explain the “paradoxical” changes in Purkinje cell firing during VOR cancellation. Thus the internal efference copy feedback can be weak when there is visual feedback through the environment that helps to maintain appropriate eye movements to stabilize the visual target.

With respect to the additional cited papers above, we note that we do not explicitly model either climbing fiber-mediated plasticity, i.e. feedback for learning, in this paper (which is the form of cerebellar plasticity described in the above models), nor feedback from the cerebellum to the cortex, which aids in motor learning and dynamics, respectively, in Boven et al. 2023 and Pemberton et al. 2022. We have added a section addressing this to the Discussion, last paragraph of section 3.4.1.

2. There are some potential limitations of the conclusions drawn due to the model inference methods used. The methods used (fmincon) can easily get stuck in local minima and more importantly they do not provide an overview of the likelihood of parameters given the data. A few studies have now shown that it is important to apply more powerful inference techniques both to infer plasticity (Costa et al. Frontiers 2013, Bykowska et al. Frontiers 2019) and neural dynamics (Gonçalves et al. eLife 2020). As highlighted by Costa et al. Frontiers 2013 using more standard fitting methods can easily lead to misleading "best parameters". Maybe this is not an issue with the present model, but it would be important to discuss why and ideally show that irrespective of the initial conditions of the model parameters (i.e. before fitting) the fitting method always produces very similar best-fit parameters.

The reviewer correctly points out that we used a deterministic model-fitting procedure. To address this concern, we complemented the full dynamic model with a simple analytic model (Figure 5) for which we could fully derive the cost function landscape and analytically show that there is a line of parameters corresponding to a perfect degeneracy in the model. Thus, the challenge in the model we analyze is that there are too many solutions, rather than it being difficult to find a solution. Given this degeneracy, we chose to fix the level of efference copy feedback and then find the (now non-degenerate) solutions, and to then compare these different solutions with regards to their implications for the correlated strengths and changes in strengths of different pathways. We have edited the relevant section of the Discussion for clarity on this topic, and have added references to the additional strategies for model inference mentioned above, in Section 3.3 “Relation to other sloppy models”.

3. There is a lack of clarity on how to best interpret inhibitory/excitatory pathways. For example, should we interpret the efference copy feedback as inhibiting or exciting PC activity? How would this pathway be implemented in the brain? The pathway is only briefly described in section 2.1 amongst the other filters; more emphasis should be placed on it to help guide the reader. You could also mention kpe at the start of section 2.2 to make it clear what it is.

We interpret this comment as referring to the questions of (1) whether our model includes a pathway for learning through feedback, (2) what is the anatomical implementation of the efference copy feedback pathway and visual pathways, and (3) how should the positive weights on the efference copy feedback pathway k _PE_ be interpreted. We address these below.

Feedback for learning was discussed in point 1 above.Anatomical implementation of efference copy pathway: We have edited the Discussion to clarify that there is anatomical evidence for efference copy input to the cerebellum, but that a key aspect of ‘feedback’ is that activity functionally loops back onto itself. Instead, neurons carrying eye movement commands (such as in the vestibular nucleus) could send signals to the cerebellum, without receiving output from the same cerebellar neurons – this would correspond to a ‘spiraling’ pathway that does not form a closed feedback loop (Figure 8). Thus we argue that the existence of the gross anatomical pathways does not necessitate a role for strong, functional, efference copy feedback (Discussion, Section 3.1, lines 481-491).

Anatomical implementation of visual pathway: The visual feedback pathways considered here are those that would receive visual motion information from the environment. This visual feedback is itself changed by eye movements, thus providing a net overall negative feedback loop that helps to stabilize gaze. This pathway has been proposed to involve cortical regions such as MST (discussed in Materials and methods, Model Implementation, lines 769-774).

Interpretation of positive feedback loop: In our model, the efference copy feedback filter, k _PE_ , has positive weight. This corresponds to the positive net sign of the Purkinje cell to brainstem to Purkinje cell feedback loop. Specifically, the Purkinje cell to brainstem pathway is inhibitory (because Purkinje cells are inhibitory), the brainstem to eye velocity command pathway is inhibitory (to achieve counter-rotation of the eyes in response to head turns), and the feedback of this eye velocity command back to Purkinje cells (k _PE_) is positive. Thus this loop in our model represents positive feedback. This is now clarified in Materials and methods, Model Implementation, lines 748.

The efference copy feedback is a net positive feedback loop in which ipsiversive head velocity excites Purkinje cells. And thank you for pointing out our lack of clarity: we have now added explicit references to k _PE_ at the end of Section 2.1 (line 138) and the start of Section 2.2 (line 153) to clarify the fact that the efference copy feedback strength was controlled in the model by setting the strength of this pathway.

4. The closed-form analysis in section 8.4 is useful but super intuitive to me. Why is there a degeneracy in how the Purkinje cell inputs are shaped and not in the brainstem inputs? Would it be possible to provide a more accessible explanation in the text? For example, I find the right plot in figure 6D surprising.

We have expanded the description of the model analysis in Materials and methods, Section 8.4 “Model Analysis”, with an additional paragraph to give more intuition for why there is a degeneracy in the Purkinje cell inputs but not the brainstem:

“We give more intuition for this analysis here. As described in the main text, the degenerate direction illustrated in Figure 5B indicates that the small Purkinje cell responses observed during the VOR in the dark could reflect either a small vestibular input (kPH) alone, or a large vestibular input offset by a large efference copy feedback (kPE). However, the full analysis here shows that the degeneracy is actually between all three input pathways to Purkinje cells (vestibular, efference copy, and visual). This degeneracy arises because there are three “unknowns” (the three input filters to Purkinje cells) but only two independently controllable variables: the vestibular and visual target stimuli. Eye velocity is determined by how these inputs are processed through the circuit; thus, it is not an independently controllable variable. In contrast, there are only two unknown input filters to the brainstem, and thus they are fully constrained by the two independently controllable variables alone.”

5. It is unclear to me *why* feedback might be important for the brain from a functional point of view (i.e. what are the advantages of no feedback vs strong feedback)? I think the paper would benefit from more discussion on this point, even if only from a more speculative angle.

Related to point 1 above, it is certainly the case that feedback is necessary for learning. We do not explicitly model the climbing fiber feedback thought to be involved in learning/plasticity of the parallel fiber pathway.

We instead focus on the role of efference copy feedback, and how it functionally impacts the required sites and signs of plasticity in the circuit. As shown in the paper, if the efference copy pathway is strong, then this is most consistent with learned changes in eye movements being driven primarily by plasticity in the brainstem pathway (as in the Miles-Lisberger hypothesis), whereas if the efference copy pathway is weak, then this is most consistent with learned changes in eye movements being driven by net depression in the parallel fiber to Purkinje cell pathway (as in the classic Marr-Albus-Ito model and as suggested by most cellular and molecular studies of parallel fiber-Purkinje cell plasticity), in addition to a role of plasticity in the brainstem pathway. We also note that, in the ‘Strong Feedback’ model, the feedback is so strong that the system is on the brink of instability – this has been argued to have the functional benefit of providing ‘inertia’ to eye movements that could help to maintain eye movements during smooth pursuit when a target goes behind an occluder, but it also has the disadvantage of placing the system at a level of positive feedback near the brink of instability. We also note that the visual feedback pathway through the environment, emphasized in this work, serves as a negative feedback loop that reduces deviations between the eye and target velocity. We have extensively re-written the first section of the Discussion (Section 3.1), in order to more clearly lay out the implications of each model for circuit plasticity and feedback.